# Cerebellar climbing fibers impact experience-dependent plasticity in the mouse primary somatosensory cortex

Abby Silbaugh[1], Kevin P Koster[2], Christian Hansel[2]*

[1]Department of Neurology, University of Chicago, Chicago, United States; [2]Department of Neurobiology and Neuroscience Institute, University of Chicago, Chicago, United States

## eLife Assessment

This study presents a **fundamental** discovery of how cerebellar climbing fibers modulate plastic changes in the somatosensory cortex by identifying both the responsible cortical circuit and the anatomical pathways. The evidence supporting the conclusions is **convincing** and well supported by modern neuroscience methodologies. Overall, this work represents a significant contribution that will be of broad interest to neuroscientists, especially those studying the long-distance cerebellar influence on non-motor brain functions.

**\*For correspondence:**
chansel@bsd.uchicago.edu

**Competing interest:** The authors declare that no competing interests exist.

**Abstract** In the cerebellum, climbing fibers (CFs) provide instructive signals for supervised learning at parallel fiber to Purkinje cell synapses. It has not been tested so far whether CF signaling may also influence plasticity in other brain areas. Here, we show that optogenetic CF activation suppresses potentiation of whisker responses in layer 2/3 pyramidal cells in the primary somato-sensory (S1) cortex of awake mice that is observed after repeated whisker stimulation. Using two-photon imaging and chemogenetics, we find that CFs control plasticity by modulating SST- and VIP-positive interneurons in S1 cortex. Transsynaptic labeling identifies zona incerta (ZI) to thalamic posterior medial nucleus projections as a pathway for cerebellar output reaching S1 cortex. Chemogenetic inhibition of PV-positive neurons in the ZI prevents CF co-activation effects, identifying the ZI as a critical relay. Our findings demonstrate that CFs impact sensory signal processing and plasticity in S1 cortex and thus may convey instructive signals.

## Introduction

Sensory experience and learning are associated with plasticity of multiple brain regions. Neocortical plasticity canonically depends on the activity of its direct input structures (*Doya, 2000*). For example, plasticity in the barrel field of the primary somatosensory (S1) cortex of rodents is evoked upon repeated whisker stimulation. Receptive field maps may reorganize after whisker clipping (*Diamond et al., 1993*; *Glazewski et al., 1999*; *Stern et al., 2001*) or a history of rhythmic whisker stimulation (*Delacour et al., 1990*; *Mégevand et al., 2009*; *Gambino et al., 2014*) that necessitates adaptive plasticity of synaptic weights, e.g., via long-term potentiation (LTP) and/or long-term depression (LTD) at synaptic inputs onto pyramidal neurons (*Feldman, 2000*; *Williams and Holtmaat, 2019*). The high degree of plasticity in S1 circuits enables adaptation to altered inputs (*Feldman and Brecht, 2005*). Though experience-dependent plasticity in the neocortex is highly responsive to input features, modulating influences from other brain regions may play a role.

The cerebellum is increasingly assigned functions that were previously attributed to the neocortex (*Schmahmann and Sherman, 1998*; *Stoodley and Tsai, 2021*). It projects via the thalamus to the motor cortex and non-motor areas, including the prefrontal cortex (*Middleton and Strick, 2001*; *Strick et al., 2009*), sensory, and associative areas (*Badura et al., 2018*; *Kelly et al., 2020*; *Pisano et al., 2021*). A plausible possibility is, therefore, that cerebellar output impacts cortical signaling. It has indeed been shown that modulation of cerebellar outputs influences activity in the frontal cortex (*Gao et al., 2018*), coherence of neuronal oscillations between neocortical areas (*Popa et al., 2013*; *McAfee et al., 2019*; *Lindeman et al., 2021*), and plasticity of parietal-motor cortex connections (*Goldenkoff et al., 2025*). However, the cellular mechanisms of these interactions have not been described.

Here, we ask the specific question of whether cerebellar climbing fiber (CF) signals have the capacity to impact neocortical plasticity. Our motivation to study CF signaling is rooted in its importance for normal cerebellar function, where CF-controlled plasticity represents a classic example of supervised learning in the brain (*Doya, 2000*). Depending on dendritic architecture, Purkinje cells (PCs) receive input from one or more CFs (*Busch and Hansel, 2023*) that originate in the contralateral inferior olive (IO). As originally predicted by Marr in his *Theory of Cerebellar Cortex*, CF signaling supervises plasticity at parallel fiber (PF) – PC synapses, providing *contexts* for learning (*Marr, 1969*). It was later experimentally demonstrated that PF and CF co-stimulation initiates LTD at PF synapses (*Albus, 1971*; *Ito and Yoshida, 1964*; *Ito and Kano, 1982*; *Ito et al., 1982*), and that the CF – via evoked calcium transients – tightly controls plasticity at these synapses (*Wang et al., 2000*; *Lev-Ram et al., 2002*; *Coesmans et al., 2004*; *Piochon et al., 2016*; *Suvrathan et al., 2016*; *Titley et al., 2019*) and provides instructive signals for cerebellum-controlled learning (*Medina et al., 2002*; *Yang and Lisberger, 2013*; *Yang and Lisberger, 2014*; *Silva et al., 2024*). CFs provide instructive error signals (*Simpson et al., 1996*), convey sensory input across modalities (*Bosman et al., 2010*; *Herzfeld and Shadmehr, 2014*; *Najafi et al., 2014b*; *Najafi et al., 2014a*; *Gaffield et al., 2019*), and may even signal the absence of expected sensory signals (*Ohmae and Medina, 2015*). CFs may also carry reward or reward prediction-related signals (*Heffley and Hull, 2019*; *Kostadinov et al., 2019*). Contexts in which CFs are recruited continue to be identified, underscoring the importance of establishing whether they may influence neocortical activity and plasticity.

To determine whether CF signaling can impact the plasticity of S1 cortex neurons, we use the mouse brain to give us experimental access at the cellular level. Our observation that CF signaling regulates plasticity of L2/3 pyramidal cells in S1 cortex shows that activity in the olivo-cerebellar system has distinct consequences for input processing and plasticity in S1 cortex, and significantly expands what is known about the basic interaction between these regions. This mechanism could be used to provide signals to the neocortex that are instructive in nature, depending on the context in which CFs are recruited, similar to those signals that play a role in cerebellar supervised learning.

## Results

### Activity-dependent plasticity in S1 cortex is regulated by optogenetic CF activation

To study the effects of CF activity on S1 plasticity, we expressed GCaMP6f in neurons in S1 cortex and used two-photon microscopy in awake mice to measure responses to air puff stimulation of the whisker field in layer 2/3 (L2/3) neurons (*Figure 1A and D*; *Figure 1—source data 1*). For optogenetic CF activation, channelrhodopsin-2 (ChR2) was expressed in neurons of the inferior olive (IO), which give rise to CFs terminating in the contralateral cerebellar cortex (*Figure 1B and E*; *Figure 1—figure supplement 1A–B*; *Figure 1—figure supplement 1—source data 1*). An LED was used to deliver 470 nm light pulses over an optical window centered on crus I/II, an area of the cerebellar cortex responsive to stimulation of the ipsilateral whisker field (*Bosman et al., 2010*). The efficacy of optogenetic activation was tested by expressing GCaMP6f in PCs of crus I/II and measuring calcium transients evoked by LED light pulses, which we varied in frequency and duration to achieve optogenetically evoked signals resembling spontaneous CF-induced calcium transients (*Figure 1C*; *Figure 1—figure supplement 1F–I*). Single light pulses lasting 50 ms evoked calcium transients which were not significantly different in amplitude or peak latency from spontaneous CF-induced events recorded in the same PCs (*Figure 1—figure supplement 1H–I*; *Simmons et al., 2022*). For plasticity

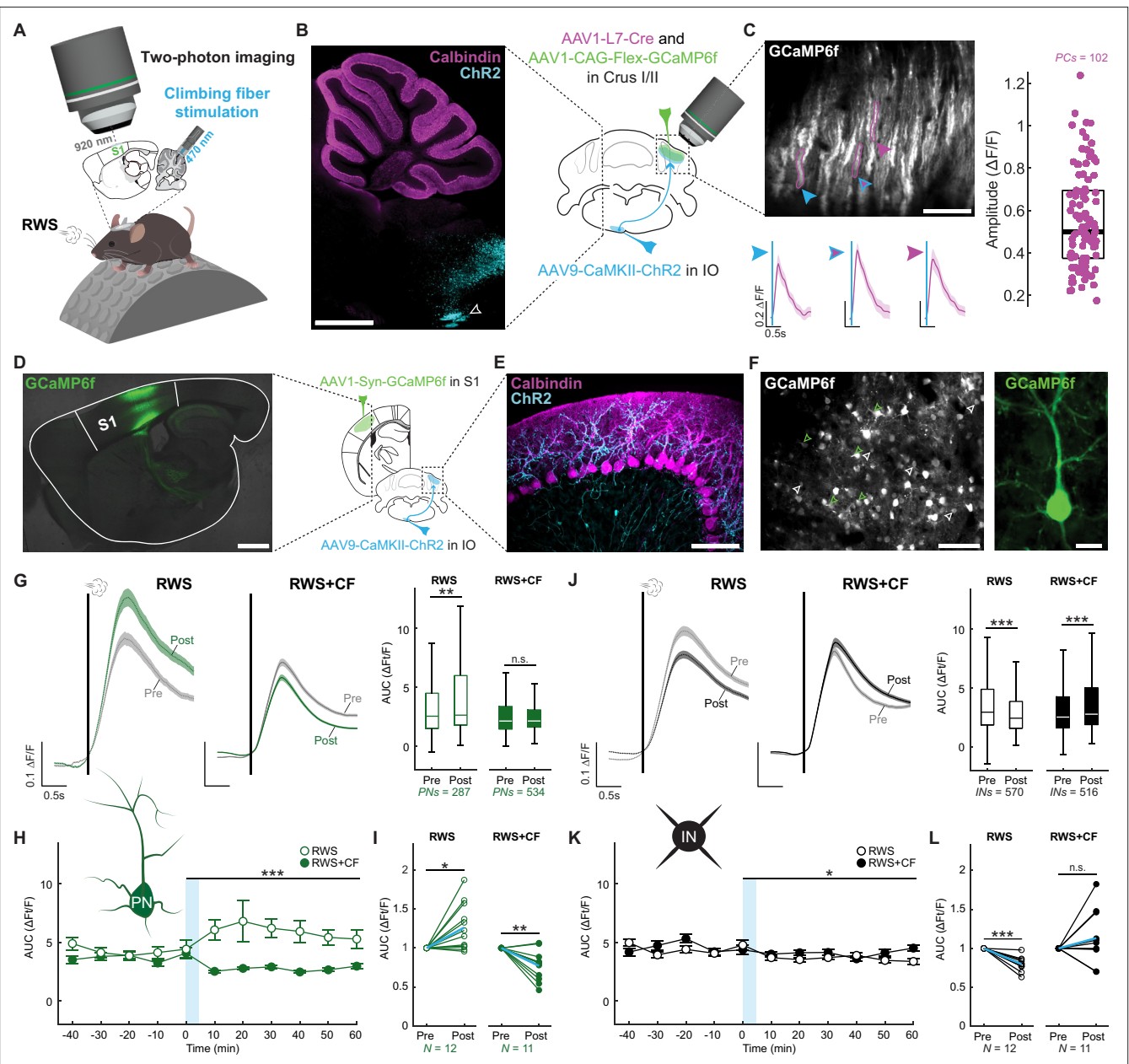

**Figure 1.** Optogenetic climbing fiber (CF) co-activation prevents adaptive L2/3 pyramidal cell potentiation in S1 cortex. (**A**) Experimental schematic for **D–L**. (**B** and **C**) ChR2 expression in the inferior olive (IO) and optogenetic CF activation evokes calcium transients in Purkinje cells (PCs). (**B**) Calbindin staining of PCs marks the cerebellum (shown here in the same plane as the IO, where ChR2 is injected; white arrowhead). Scale bar: 1 mm. (**C**) Two-photon field of view of PC dendrites with calcium responses evoked by 470 nm, 50 ms LED pulses. Scale bar: 100 μm. Right: Distribution of response amplitudes (box: 25th and 75th percentiles; line: median; points: mean across trials). Trace scale bar: 0.2 ΔF/F; 0.5s; line represents mean; shaded region represents SEM across neurons. (**D**) GCaMP6f expression in S1 cortex. Scale bar: 1 mm. (**E**) Sagittal view of ChR2-expressing CF terminals in the contralateral cerebellar cortex that are activated in optogenetic experiments. Scale bar: 100 μm. (**F**) Left: example of the morphological identification of L2/3 pyramidal cells by apical dendrites (green arrowheads) and INs (white arrowheads) in a two-photon field of view. Scale bar: 100 μm. Right: two-photon field of view of Pyramidal cell at higher magnification. Scale bar: 15 μm. (**G**) Pyramidal cell plasticity after RWS (N=12 mice, n=287 neurons) and after pairing with CF activation (RWS+CF; N=11 mice, n=534 neurons). Trace scale bars: 0.1 ΔF/F; 0.5 s; line represents mean; shaded region represents SEM across neurons. Right: quantification of the evoked responses across neurons. Box: 25th and 75th percentiles; line: median; whiskers: range. (**H**) Data in **G** binned across time. Blue shading: RWS or RWS+CF. Points: mean; error bars: SEM across neurons. (**I**) Quantification across mice, normalized to the pre-plasticity response. Blue line: mean across animals. (**J–L**) Analyses for INs in the same mice; data visualization as in **G-I**. All summary data, statistical methods, and significance levels are available in *Figure 1—source data 1*.

The online version of this article includes the following source data and figure supplement(s) for figure 1:

*Figure 1 continued on next page*

Figure 1 continued

**Source data 1.** Summary data, statistical methods, and significance levels for data in *Figure 1*.

**Figure supplement 1.** Optogenetic climbing fiber (CF) activation causes GCaMP6f-encoded calcium transients in Purkinje cells.

**Figure supplement 1—source data 1.** Summary data, statistical methods, and significance levels for data in *Figure 1—figure supplement 1*.

**Figure supplement 2.** Plasticity in different neuron types.

**Figure supplement 2—source data 1.** Summary data, statistical methods, and significance levels for data in *Figure 1—figure supplement 2*.

**Figure supplement 3.** Impact of repeated whisker stimulation (RWS)- and RWS + climbing fiber (RWS+CF)-stimulation on animal motion.

**Figure supplement 3—source data 1.** Summary data, statistical methods, and significance levels for data in *Figure 1—figure supplement 3*.

**Figure supplement 4.** Intrinsic optical imaging confirms climbing fiber (CF)-mediated suppression of S1 plasticity.

**Figure supplement 4—source data 1.** Summary data, statistical methods, and significance levels for data in *Figure 1—figure supplement 4*.

induction in S1 cortex, we used a multi-whisker stimulation protocol ('rhythmic whisker stimulation;' RWS) similar to those previously used to study sensory-driven synaptic plasticity in vivo (*Gambino et al., 2014*; *Williams and Holtmaat, 2019*; *Jouhanneau et al., 2014*; *Audette et al., 2019*). Here, RWS consisted of an application of 100 ms air puffs (8 psi) to the contralateral whisker field for 5 min at 8 Hz, a natural whisking frequency for mice that are sampling the environment (*Diamond et al., 2008*). RWS stimulation resulted in an increase in whisker-evoked calcium responses (ΔF/F over the range of 0–700 ms post-stimulus onset) that we observed in morphologically identified L2/3 pyramidal neurons (PNs; *Figure 1F*) for the duration of the recording (60 min post-RWS; *Figure 1G–I*; *Figure 1—figure supplement 2D*; *Figure 1—figure supplement 2—source data 1*). These somatic calcium responses measured with GCaMP6 correlate with neuronal spike rates (*Theis et al., 2016*). L2/3 pyramidal neurons respond to stimulation of the corresponding principal whisker but may also respond to stimulation of surround whiskers (*Brecht et al., 2003*). Therefore, the plasticity observed here in response to multi-whisker stimulation may reflect both a potentiation of synapses already conveying principal and surround whisker input as well as an expansion of the neuron's receptive field to include additional surround whiskers. Given that increases in the activity of excitatory neurons may be coupled with decreased activity of local inhibitory neurons in sensory-evoked plasticity (*Heiss et al., 2008*), we also assessed the responses of neurons that were not identified as pyramidal cells (i.e. putative interneurons; INs) in the same mice. Indeed, INs collectively responded to RWS stimulation with a significant depression in the amplitude of evoked calcium events (*Figure 1J–L*; *Figure 1—figure supplement 2E*). As parvalbumin (PV)-positive interneurons, somatostatin (SST)-positive interneurons, and vasoactive intestinal polypeptide (VIP)-positive interneurons make up ~40%, ~30%, and ~12% of the neocortical interneuron population (*Staiger and Petersen, 2021*; *Tremblay et al., 2016*), respectively, this plasticity effect is likely comprised of a substantial portion of SST and PV interneurons. Using transgenic mice to tag SST and PV interneurons with tdTomato, we demonstrate RWS causes a significant depression in SST and PV interneurons (*Figure 1—figure supplement 2F and G*) – consistent with our observations in putative interneurons.

To determine whether CF co-activation with whisker stimulation influences this form of experience-dependent plasticity, 50 ms light pulses were delivered at 1 Hz during the 5 min period of repeated whisker stimulation (RWS+CF), which is the rate of spontaneous CF discharge (*Simpson et al., 1996*). Each light pulse was delivered with a delay of 45 ms with respect to the onset of whisker stimuli, mimicking the natural latency of CF responses to whisker stimulation (*Bosman et al., 2010*). RWS+CF blocked the potentiation of L2/3 pyramidal cell responses in S1 cortex and depression was observed instead (*Figure 1G–I*; *Figure 1—figure supplement 2D*). The depression of IN responses observed after RWS was absent upon optogenetic CF co-activation (*Figure 1J-L*, *Figure 1—figure supplement 2E*), and recordings in mice with tdTomato-tagged SST and PV interneurons further demonstrated the absence of inhibitory interneuron depression (*Figure 1—figure supplement 2F–G*). Importantly, both the S1 cortex plasticity and optogenetic CF co-activation effects were observed when trials with active movement of the mouse were included (*Figure 1*; *Figure 1—figure supplement 2*) or when analysis was restricted to trials during which the mice were resting both before stimulus onset and throughout the entire response period (*Figure 1—figure supplement 3*; *Figure 1—figure supplement 3—source data 1*). Note movements of the vibrissae were absent during rest.

To test the robustness of these effects, we reproduced these findings using an alternate methodology that is well-suited to measure receptive fields, including barrels in S1 cortex. Intrinsic optical imaging (*Bonhoeffer and Grinvald, 1991*) was used to measure plasticity effects at low spatial resolution in anesthetized mice to facilitate measurement of single whisker responses. We slipped a glass pipette over an individual untrimmed whisker and moved it back and forth at 8 Hz for a period of 5 min. In the barrel field of contralateral S1 cortex, we confirmed the activation of the corresponding barrel and recorded responses to this passive sensory experience for 20 min before and 30 min after repeated stimulation. We observed an increase in the responsive area beyond the barrel (*Figure 1—figure supplement 4*; *Figure 1—figure supplement 4—source data 1*). Plasticity was not observed in S1 cortex when ChR2-expressing CFs in crus I/II of the cerebellum were optogenetically activated at 1 Hz for 5 min during the period of repeated stimulation or when CFs alone were optogenetically activated at 1 Hz for 5 min (*Figure 1—figure supplement 4*). Thus, intrinsic optical imaging demonstrated that sensory-evoked receptive field plasticity of barrels can be regulated by cerebellar signaling.

Taken together, our findings demonstrate that whisker-induced response potentiation in S1 cortex results from L2/3 pyramidal cell plasticity. Local inhibitory interneurons show the opposite direction of plasticity in the same animals. The observation that this IN plasticity is of lower amplitude and does not perfectly mirror pyramidal cell plasticity suggests that the former supports the latter and may help to control it, but that pyramidal cell LTP may take place even in the absence of LTD in INs. Note that 'LTP' and 'LTD' here reflect target neuron responsiveness – as measured by neural calcium signals – and are not specifically shown to rest on synaptic gain changes. CF activity blocks the whisker-induced response potentiation in PNs and concomitant depression in INs, thus regulating this form of whisker map plasticity.

## CF activation modulates inhibitory interneurons in S1 cortex

Local inhibitory interneurons in the neocortex are known to gate plasticity of pyramidal neurons, a mechanism studied extensively in the rodent barrel cortex (*Tremblay et al., 2016*; *Williams and Holtmaat, 2019*). Given our observation that CF activation exerts a gating effect on pyramidal neuron plasticity, it is plausible that neocortical inhibitory interneurons ultimately mediate the effects of CF signaling on S1 plasticity. To test this, we first examined whether CF activation causes acute modulation of the responses of inhibitory interneurons to whisker stimulation (*Figure 2A*; *Figure 2—source data 1*). Paired whisker and CF stimulation indeed caused an acute increase in the whisker-evoked response of INs (*Figure 2C*). In the same mice, CF co-stimulation caused a concomitant decrease in the whisker-evoked response of PNs (*Figure 2B*). The acute response increase observed in the nonspecific population of INs here is likely driven by SST and PV neurons, which together make up ~70% of the S1 inhibitory interneuron population (*Tremblay et al., 2016*). As such, we further characterized the responses of tdTomato-tagged SST and PV neurons (*Figure 2E–F*). Matching the results of the IN population, the acute responses of SST and PV neurons to whisker stimulation were enhanced upon CF co-activation (*Figure 2H–I*). VIP interneurons – which primarily inhibit other interneurons, including SSTs and PVs (*Tremblay et al., 2016*) – showed no consistent change in their acute responses to whisker stimulation upon CF co-activation in the default analysis window (0–700 ms). However, VIPs demonstrated significant suppression in a later analysis window of 650–850 ms (corresponding to the early component of the post-peak response; *Figure 2G*). These findings suggest that CF co-activation causes a range of response changes in the population of L2/3 VIP neurons, but that an overall suppressive effect emerges in the late response phase. Taken together, these results demonstrate that CFs have the capacity to regulate the activity of excitatory and inhibitory neurons in S1 by differentially modulating their acute responses to sensory input.

## Opposing roles of SST and VIP interneurons in CF-mediated control of S1 plasticity

SST interneurons contact the dendrites of L2/3 pyramidal cells (*Figure 3A and E*; *Figure 3—source data 1*), where inhibition can impact local synaptic plasticity processes. As we observed that CF co-activation with whisker stimulation causes an acute increase in the activity of SST interneurons, we tested their necessity and sufficiency in mediating the effects of CF activity on S1 plasticity using chemogenetic approaches. We first expressed the excitatory hM3D(Gq) DREADD in neocortical SST interneurons (*Figure 3B*). Administration of the DREADD agonist deschloroclozapine (DCZ) by intraperitoneal

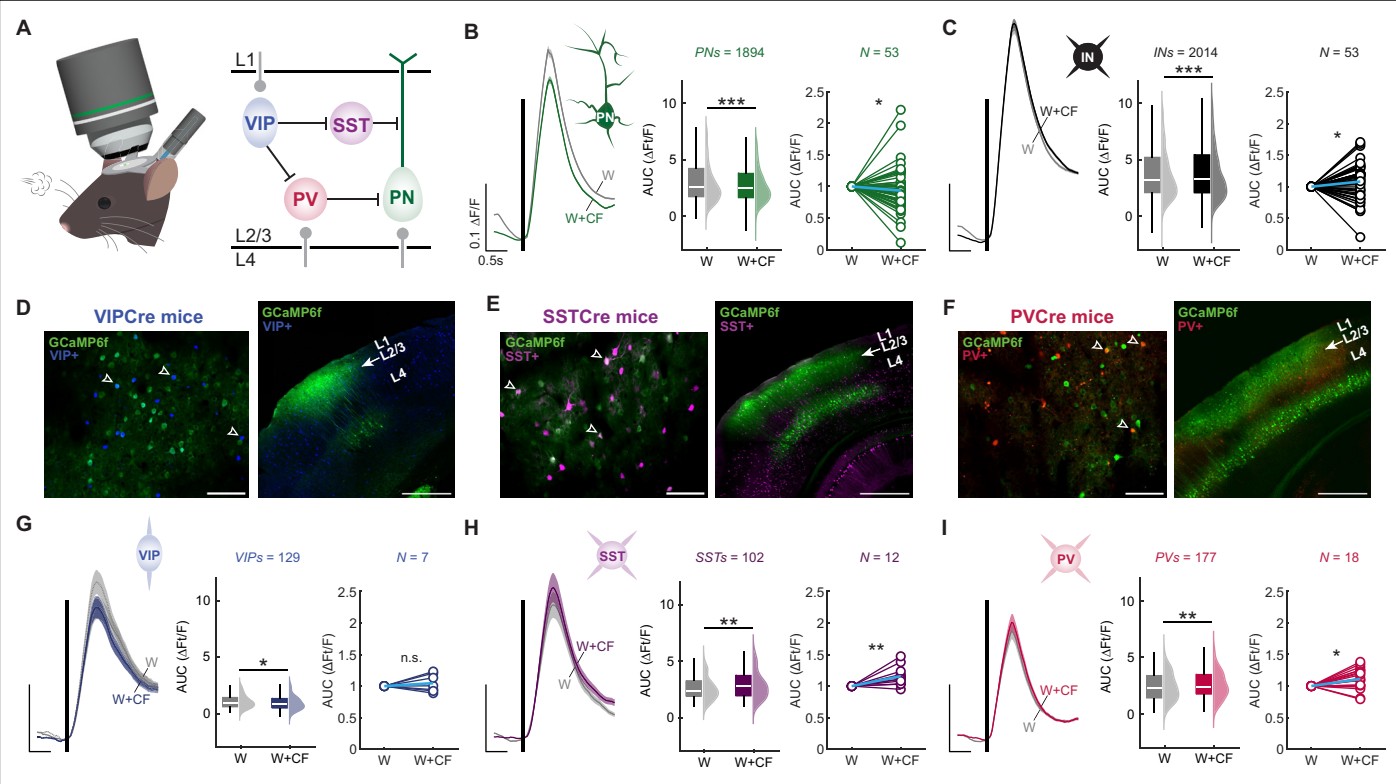

**Figure 2.** Optogenetic climbing fiber (CF) co-activation differentially modulates basic responses to whisker stimulation in different types of neocortical neurons. (**A**) Schematic of the recording configuration (left) and S1 cortex microcircuit (right). (**B**) CF co-activation with whisker stimulation significantly alters evoked responses of morphologically identified L2/3 pyramidal neurons. Left: averaged traces. Trace scale bars: 0.1 ΔF/F; 0.5 s; shaded region: SEM across neurons. Middle: analysis by neuron (AUC from 0 to 700 ms). Box: 25th and 75th percentiles; line: median; whiskers: range. Violin plot: kernel density. Right: analysis by mouse, normalized to whisker-only response. Blue line indicates mean across animals. (**C**) CF co-activation significantly enhances responses in putative interneurons (interneurons INs; n=2014 cells; N=53 mice) in the same mice. (**D–F**) Sample two-photon field of view (left) and post-hoc verification of expression in S1 using confocal microscopy (right). Arrowheads indicate neurons co-expressing GCaMP6f and tdTomato in vasoactive intestinal polypeptide (VIP) interneurons (**D**), somatostatin (SST) interneurons (**E**), or parvalbumin (PV) interneurons (**F**). Scale bars: 100 µm (left); 500 µm (right). (**G**) CF co-activation with whisker stimulation significantly reduces VIP population responses (n=129 cells, N=7 mice), but enhances responses in SST interneurons (n=102 cells; N=12 mice; **H**) and PV interneurons (n=177 cells; N=18 mice; **I**). Note VIP responses are reduced in a late analysis window (650–850 ms), while SST and PV responses are enhanced in the default 0–700 ms analysis window. Middle plots show analysis by neuron, right plots analysis by mouse (blue line indicates mean). Data visualization for **C** and **G-I** as in **B**. All summary data, statistical methods, and significance levels are available in *Figure 2—source data 1*.

The online version of this article includes the following source data for figure 2:

**Source data 1.** Summary data, statistical methods, and significance levels for data in *Figure 2*.

injection caused an increase in SST interneuron responses (*Figure 3C*). Under these conditions, RWS did not potentiate L2/3 pyramidal cell responses, and calcium signals were depressed instead (*Figure 3D*), demonstrating an increase in SST activity is sufficient to mimic the suppressive effect of CF signaling on PNs (*Figure 1G–I*). In contrast, inhibitory hM4D(Gi) DREADD expression in SST interneurons (*Figure 3F*) and acute administration of DCZ caused a significant reduction in SST interneuron responses (*Figure 3G*). This rescued potentiation of L2/3 pyramidal cell responses when RWS was paired with CF co-activation (*Figure 3H*), demonstrating SST activity is required to mediate the suppressive effect of CF signaling on PNs. These findings confirm that neocortical SST interneurons are directly positioned to control plasticity of L2/3 pyramidal neurons (*Williams and Holtmaat, 2019*) and suggest that this mechanism is recruited by CF co-activation.

SST interneurons receive relatively weak input from cortical and subcortical regions, providing input to S1 but are strongly inhibited by local VIP interneurons (*Figure 4A and E*; *Figure 4—source data 1*; *Sermet et al., 2019*; *Audette et al., 2018*). Activation of VIP interneurons may suppress SST activity during RWS, thus enabling pyramidal neuron potentiation (*Williams and Holtmaat, 2019*;

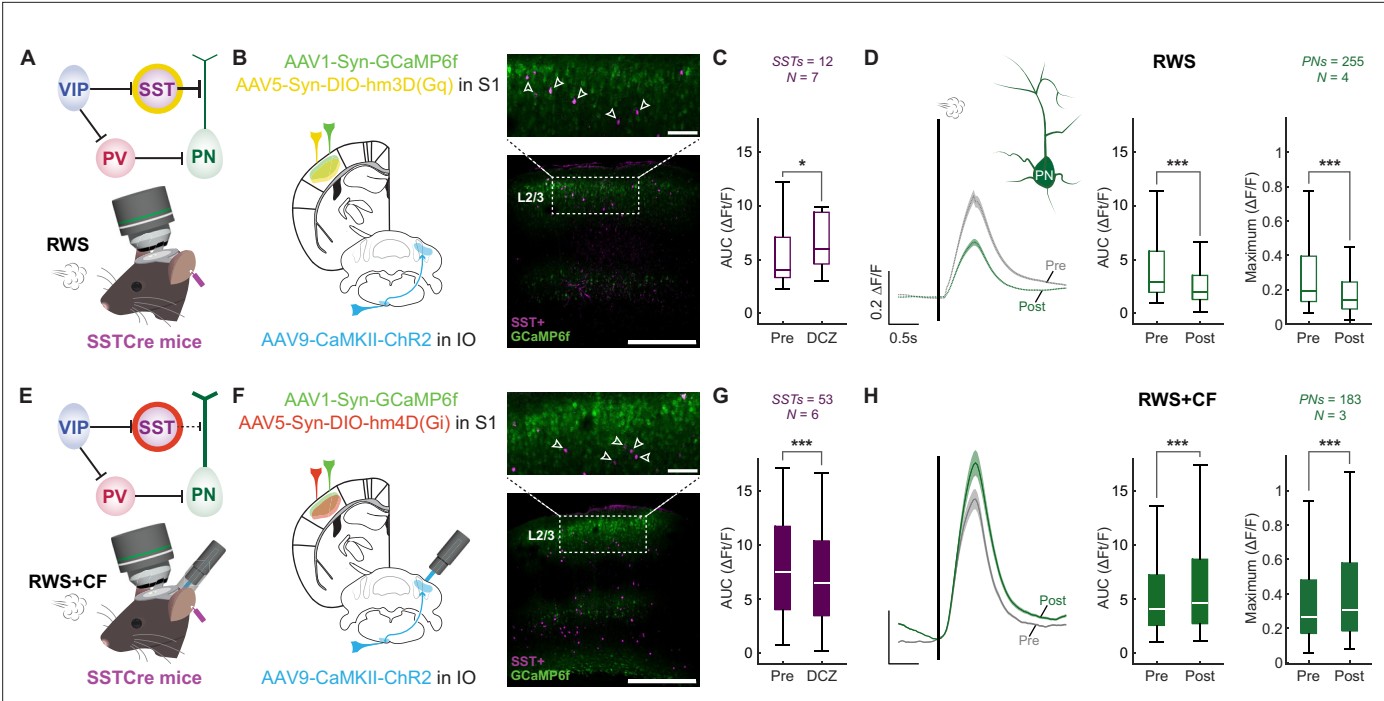

**Figure 3.** Somatostatin (SST) interneurons are recruited to prevent L2/3 pyramidal cell plasticity. (**A**) Circuit diagram highlighting the chemogenetic activation of SST interneurons and recording configuration. (**B**) Expression of activating hM3D(Gq) receptors is localized to neocortical SST interneurons. Scale bar: 100 µm (top) and 500 µm (bottom). (**C**) Deschlorociozapine (DCZ) application enhances responses in SST interneurons (n=12 cells; N=7 mice). Box: 25th and 75th percentiles; line: median; whiskers: range. (**D**) In the presence of DCZ, repeated whisker stimulation (RWS) stimulation suppresses L2/3 pyramidal cell potentiation, and even causes depression, even when the climbing fiber (CF) is not co-activated (n=255 cells; N=4 mice). Left: average traces; middle: AUC analysis by cell; right: maximum response amplitude by cell. Trace scale bars: 0.2 ΔF/F; 0.5 s; shaded region: SEM across neurons. Box: 25th and 75th percentiles; line: median; whiskers: range. (**E**) Circuit diagram highlighting the chemogenetic inhibition of SST interneurons and recording configuration. (**F**) Expression of inhibiting hM4D(Gi) receptors is localized to neocortical SST interneurons. Scale bars as in **B**. (**G**) DCZ application reduces responses in SST interneurons (n=53 cells; N=6 mice). (**H**) In the presence of DCZ, pyramidal cell potentiation is observed, despite CF-co-activation (n=183 cells; N=3 mice). Data visualization for **G-H** is the same as **C-D**. All summary data, statistical methods, and significance levels are available in *Figure 3—source data 1*.

The online version of this article includes the following source data for figure 3:

**Source data 1.** Summary data, statistical methods, and significance levels for data in *Figure 3*.

*Williams et al., 2025*). We have shown above that optogenetic CF activation causes an overall reduction in whisker-evoked responses of VIP interneurons (*Figure 2G*), which may be a mechanism through which CFs block pyramidal neuron potentiation. To examine whether this VIP activity regulation is critical for RWS+CF-induced suppression of L2/3 pyramidal cell plasticity, we used chemogenetic approaches to manipulate the activity of VIP interneurons. We first observed that expression of inhibitory hM4D (Gi) DREADDs in VIP interneurons (*Figure 4B*) and acute DCZ administration significantly reduced activity in VIP interneurons (*Figure 4C*) and prevented S1 potentiation upon RWS stimulation alone (*Figure 4D*). This manipulation was similar to the effect of RWS+CF (*Figure 1G–I*) or SST activation during RWS (*Figure 3D*). Conversely, expression of excitatory hM3D(Gq) DREADDs in VIP interneurons (*Figure 4F*) and acute DCZ administration enhanced VIP interneuron responses (*Figure 4G*) and rescued S1 plasticity upon paired RWS and optogenetic CF activation (RWS+CF; *Figure 4H*), mimicking the effect of RWS (*Figure 1G–I*) or suppression of SST interneuron activity during RWS+CF (*Figure 3H*).

To control for the continued presence of DCZ, identical experiments for each condition were performed without the administration of RWS (*Figure 4—figure supplement 1A and D*; *Figure 4—figure supplement 1—source data 1*) or RWS+CF (*Figure 4—figure supplement 1G, J and M*). When comparing control and plasticity experiments, both the SST activation and VIP inactivation experiments successfully show SST activation is sufficient to block RWS-mediated PN potentiation (*Figure 4—figure supplement 1C*), and VIP activity is required for RWS-mediated PN potentiation

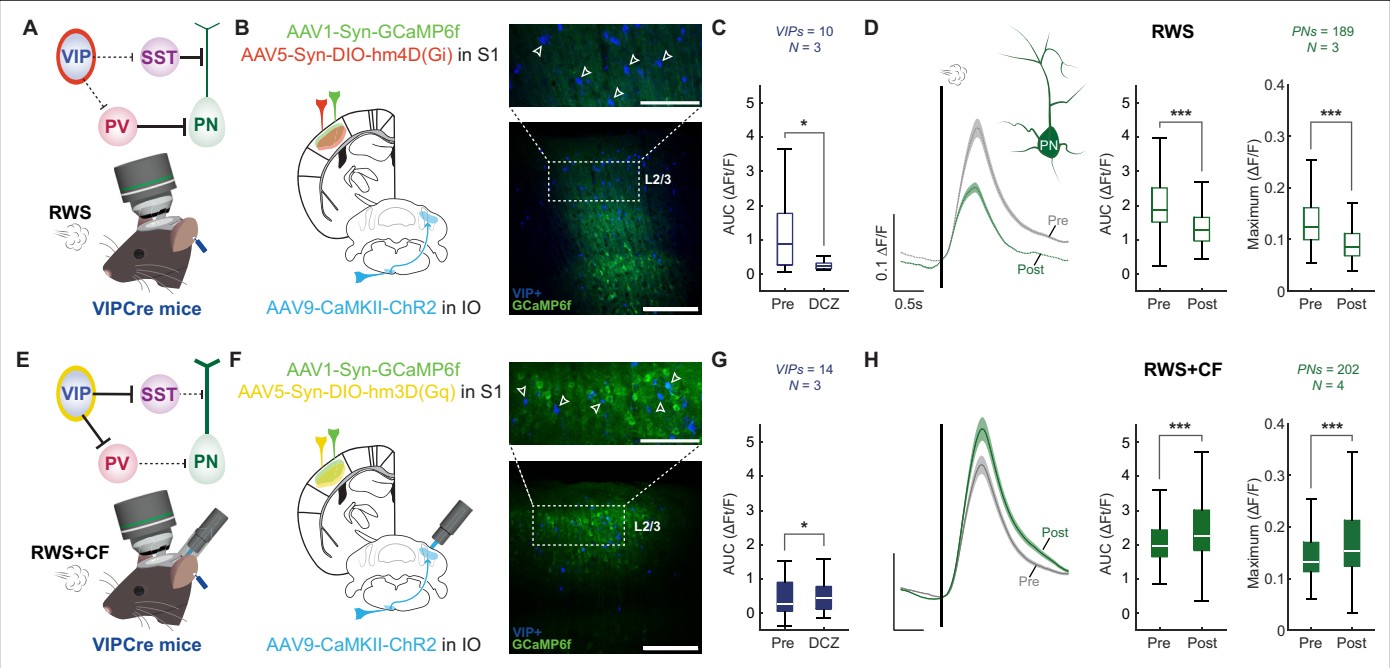

**Figure 4.** Vasoactive intestinal polypeptide (VIP) interneurons mediate the effects of optogenetic climbing fiber (CF) activation. (**A**) Circuit diagram highlighting the chemogenetic inhibition of VIP interneurons and recording configuration. (**B**) Expression of inhibiting hM4D(Gi) receptors is localized to neocortical VIP interneurons. Scale bars: 100 µm (top) and 500 µm (bottom). (**C**) Deschloroclozapine (DCZ) application reduces responses in VIP interneurons (n=10 cells; N=3 mice). Box: 25th and 75th percentiles; line: median; whiskers: range. (**D**) In the presence of DCZ, repeated whisker stimulation (RWS) stimulation suppresses L2/3 pyramidal cell potentiation, and even causes depression, even when the CF is not co-activated (n=189 cells; N=3 mice). Left: averaged traces; middle: AUC analysis by cell; right: maximum response amplitude by cell. Trace scale bars: 0.1 ΔF/F; 0.5 s; shaded region: SEM across neurons. Box: 25th and 75th percentiles; line: median; whiskers: range. (**E**) Circuit diagram highlighting the chemogenetic activation of VIP interneurons and recording configuration. (**F**) Expression of activating hM3D(Gq) receptors is localized to neocortical VIP interneurons. Scale bars as in **B**. (**G**) DCZ application enhances responses in VIP interneurons (n=14 cells; N=3 mice). (**H**) In the presence of DCZ, pyramidal cell potentiation is observed, despite CF co-activation (n=202 cells; N=4 mice). Data visualization for **G-H** is the same as **C-D**. All summary data, statistical methods, and significance levels are available in *Figure 4—source data 1*.

The online version of this article includes the following source data and figure supplement(s) for figure 4:

**Source data 1.** Summary data, statistical methods, and significance levels for data in *Figure 4*.

**Figure supplement 1.** Plasticity effects in L2/3 pyramidal neurons (PNs) vs controls in chemogenetic manipulation experiments.

**Figure supplement 1—source data 1.** Summary data, statistical methods, and significance levels for data in *Figure 4—figure supplement 1*.

(*Figure 4—figure supplement 1F*). The VIP activation experiments also successfully showed VIP activation rescues PN potentiation after RWS+CF (*Figure 4—figure supplement 1O*) and that this effect was not due to sustained increases in responsivity of pyramidal neurons caused by the presence of DCZ in these conditions (*Figure 4—figure supplement 1N*). We additionally tested the potential contribution of PV interneurons to the CF-mediated control of S1 plasticity. Expression of inhibitory hM4D (Gi) DREADDs in PV interneurons indeed rescued PN potentiation after RWS+CF when compared to controls (*Figure 4—figure supplement 1L*), a robust effect given that PV interneurons are usually basket cells with strong recurrent connectivity to pyramidal neurons (*Tremblay et al., 2016*).

To further support our interpretation that VIP interneurons control a disinhibitory network in S1 cortex, we analyzed the IN populations (which consist of putative SSTs and PVs) in the VIP chemogenetic manipulation experiments (*Figure 5*; *Figure 5—source data 1*). VIP inactivation during RWS, which blocked PN potentiation (*Figure 4D*), indeed caused a corresponding increase in IN activity (*Figure 5C*) to mimic the effects of RWS+CF (*Figure 1G–L*) even without CF activity. VIP activation during RWS+CF, which rescued PN potentiation (*Figure 4H*), caused a corresponding decrease in IN activity (*Figure 5F*) to mimic the effects of RWS (*Figure 1G–L*) even when CFs were active. These findings confirm that VIP interneurons orchestrate the inhibitory network in S1 cortex and show that this mechanism is recruited by optogenetic CF activation. Taken together, the findings presented thus

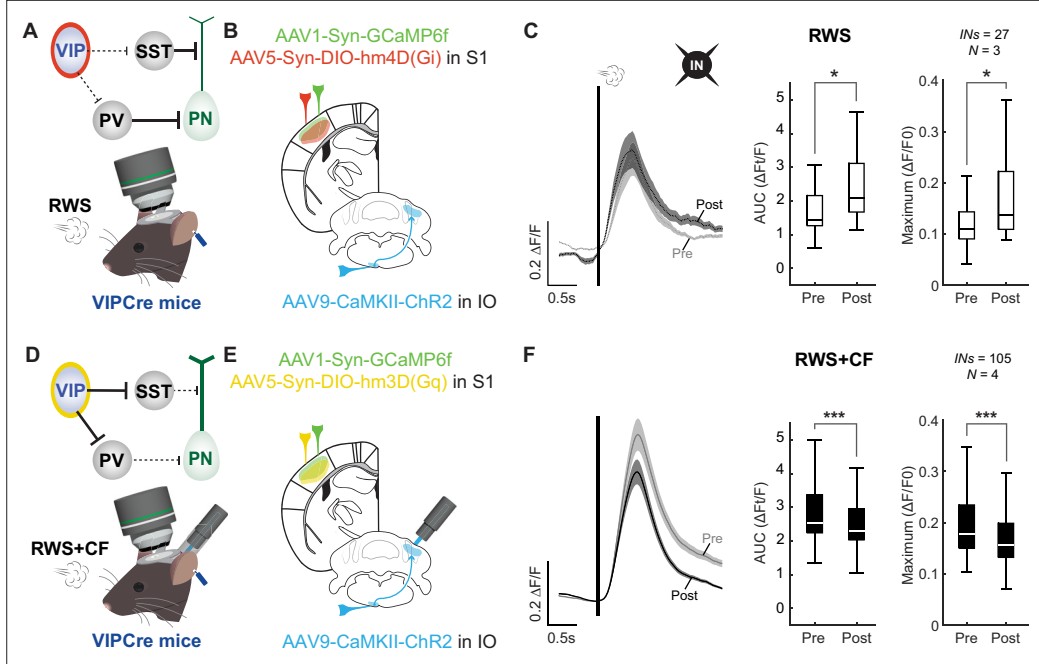

**Figure 5.** Vasoactive intestinal polypeptide (VIP) interneurons orchestrate the inhibitory network in S1. (**A**) Circuit diagram highlighting the chemogenetic inhibition of VIP interneurons and recording configuration. (**B**) Inhibiting hM4D(Gi) receptors are expressed in neocortical VIP interneurons. Note A-B are identical to *Figure 4A–B*. (**C**) In the presence of deschloroclozapine (DCZ), repeated whisker stimulation (RWS) stimulation activates L2/3 interneurons, even when the climbing fiber (CF) is not co-activated (n=27 cells; N=3 mice). Left: averaged traces; middle: AUC analysis by cell; right: maximum response amplitude by cell. Trace scale bars: 0.2 ΔF/F; 0.5 s; shaded region: SEM across neurons. Box: 25th and 75th percentiles; line: median; whiskers: range. (**D**) Circuit diagram highlighting the chemogenetic activation of VIP interneurons and recording configuration. (**E**) Activating hM3D(Gq) receptors are expressed in neocortical VIP interneurons. Note D-E are identical to *Figure 4E–F*. (**F**) In the presence of DCZ, IN depression is observed, despite CF co-activation (n=105 cells; N=4 mice). Data visualization is the same as in **C**. All summary data, statistical methods, and significance levels are available in *Figure 5—source data 1*.

The online version of this article includes the following source data for figure 5:

**Source data 1.** Summary data, statistical methods, and significance levels for data in *Figure 5*.

far show that the critical circuit for RWS-driven plasticity is located in S1 cortex itself and identify VIP and SST interneurons as effectors that are sufficient to mediate the consequences of CF activation. Here, the notion of sufficiency does not exclude potential effects of plasticity processes elsewhere that might well modulate effector activation in this context and others not yet tested.

## Zona incerta to the thalamic posterior medial nucleus (POm) as a multi-synaptic pathway from the cerebellar nuclei to S1 cortex

Cerebellar climbing fibers must exert their effects on the neocortex via the deep cerebellar nuclei (DCN), beyond which there may be several potential pathways to the primary somatosensory cortex. The barrel cortex in mice receives whisker sensory information from two thalamic sources, the ventral posterior medial thalamic nucleus (VPM) and the POm. Whereas the VPM projects densely to the middle layers of S1 cortex, POm inputs terminate largely in L5 as well as L1, where, in addition to pyramidal cells, they synapse on VIP interneurons (*Sermet et al., 2019*; *Audette et al., 2018*). POm-to-VIP inputs have previously been shown to gate RWS-mediated plasticity of PNs in barrel cortex (*Audette et al., 2019*; *Williams and Holtmaat, 2019*). However, because the DCN sends only weak direct projections to POm (and VPM), we focused on other output regions of the cerebellum that could influence the activity of sensory thalamic nuclei: the thalamic reticular nucleus (TRN) and zona incerta (ZI). The ZI represents a more plausible pathway through which the CF could exert its effects on S1 given its robust innervation by the cerebellar nuclei (*Pisano et al., 2021*; *Novello et al., 2024*; *Zhao et al., 2025*) and projections to the POm (*Barthó et al., 2002*). To rule out a pathway to S1

through the thalamic reticular nucleus, whose GABAergic neurons primarily inhibit VPM (*Martinez-Garcia et al., 2020*), we used retrograde fluoro-Gold labeling from the somatosensory thalamus (which included VPM and POm) and anterograde labeling from the contralateral DCN. We found no overlap in the TRN (*Figure 6—figure supplement 1*). Next, we determined the primary output pathways of ZI cells that receive cerebellar input by using a dual-injection approach which leveraged the transsynaptic transport properties of AAV1 (*Zingg et al., 2017*): AAV1-Flp recombinase (pAAV-EF1a-Flpo) was delivered to the deep cerebellar nuclei, followed by a Flp-dependent label (Frt-ChR2-EYFP) to contralateral ZI (*Figure 6A*; *Figure 6—source data 1*). Abundant EYFP-expressing neurons were found in the ZI (*Figure 6B and C*; *Figure 6—figure supplement 2*), with their axon terminals visible in both POm (*Figure 6B and C*; *Figure 6—figure supplement 2*) and S1 cortex (*Figure 6B and C*). These results suggest that the ZI-POm pathway, and not the TRN-VPM pathway, constitutes a critical thalamic route for cerebellar signals reaching S1 cortex (*Figure 6—figure supplement 2C*).

To assess whether there is a monosynaptic connection from the cerebellar nuclei to neurons in the ZI that project to POm, we expressed ChR2 in the cerebellar nuclei and performed in vitro whole-cell patch-clamp recordings from POm-projecting ZI neurons (i.e. cells that were retrogradely labeled by fluoro-Ruby injected into POm) and tested them for ChR2-evoked responses (*Figure 6D*, *Figure 6—figure supplement 3*). Optogenetic activation of ChR2+ terminals at 10 Hz elicited EPSPs that persisted in the presence of bath-applied TTX (1 μM; *Figure 6E and F*), despite significant amplitude reduction. This TTX-insensitive component represents the ChR2-evoked EPSP at the synapses between cerebellar nuclei neurons and ZI neurons and demonstrates the monosynaptic nature of this pathway (*Cho et al., 2013*).

It has previously been demonstrated that most POm-projecting neurons in ZI are PV neurons (*Trageser et al., 2006*). To test whether these neurons convey cerebellar instructive signals to S1 cortex, we chemogenetically inhibited PV neurons in ZI during basic whisker and whisker plus optogenetic CF activation conditions and measured responses in L2/3 pyramidal neurons in S1 cortex. Expression of inhibitory hM4D (Gi) DREADDs in ZI-residing PV neurons (*Figure 6G*) and DCZ application prevented the suppressive effect of CF co-activation on pyramidal cell responses in S1 cortex (*Figure 6H and I*). These data demonstrate that thalamus-projecting PV neurons in ZI convey CF signals to S1 cortex.

## Discussion

Multiple functions have been assigned to cerebellar CFs. As in our recordings, optogenetic CF co-activation suppresses S1 pyramidal neuron plasticity in an experimental paradigm lacking behavioral reward, this behaviorally neutral signal acts like an error signal would. However, it remains possible that CF recruitment and regulation of S1 plasticity may happen in a reward/reward omission context as well. CF activation might be driven by reward/reward omission in sensory-association training. This might be the reason why sensory stimulation in such training context drives cortical plasticity less well than passive sensory stimulation alone (*Zhu et al., 2024*). Plasticity in S1 cortex results from an update in sensory input – here exposure to a new sensory experience, prolonged rhythmic whisker activation at 8 Hz – that necessitates circuit adaptation. CF signaling may get recruited when an obvious disruption or violation of a sensory prediction occurs, or simply when no stability in sensory input is reached. The absence of *sameness* may be detected by inferior olive neurons as an error in the sensory environment, and subsequently, CFs prevent the otherwise unfolding potentiation. Such a function provides an example for a 'sensory role' of the olivo-cerebellum (*Bower, 1997*; *Baumann et al., 2015*), in this case by controlling adaptive plasticity in a primary sensory cortex, without immediate effects on the motor domain.

In rodents, direct trigemino-olivary connections convey whisker-related information to the olive (*Bosman et al., 2011*). Passive whisker activation evokes CF-mediated complex spike firing in PCs in ipsilateral crus I/II (*Bosman et al., 2010*). These evoked spikes occur on top of a spontaneous complex spike firing rate of ~1 Hz (*Simpson et al., 1996*). What distinguishes CF activity that suppresses plasticity in the cerebral cortex from these more common activity patterns? First, prolonged low-frequency air puff stimulation to sensitive skin areas on the snout or wrist of rodents evokes potentiation, and not depression, of PC responses (*Ramakrishnan et al., 2016*; *Lin et al., 2024*). This finding is in line with the current understanding of CF signaling: novelty activates it (*Simpson et al., 1996*); when stimuli are repeatedly presented (producing/inhabiting a state of *sameness*), evoked complex spike firing weakens. Thus, RWS stimulation likely will not produce continuous elevated CF

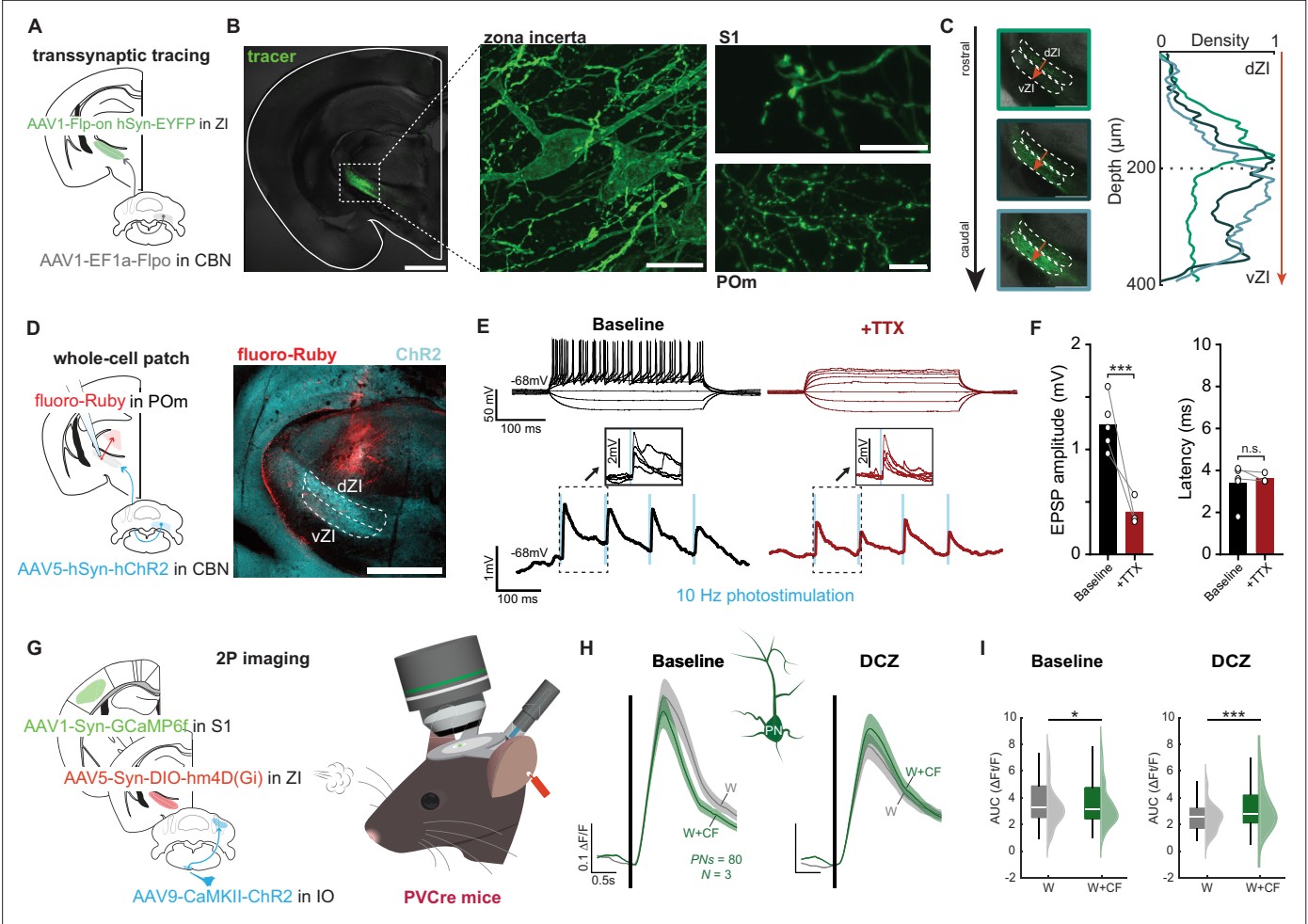

**Figure 6.** A pathway from the cerebellar nuclei via the zona incerta and thalamic nucleus posterior medial nucleus (POm) to S1 cortex. (**A**) Schematic of dual-injection strategy to label outputs of zona incerta (ZI) neurons receiving input from contralateral cerebellar nuclei (CBN). (**B**) EYFP expression in the left hemisphere (left; scale bar: 1 mm) with higher magnification images taken in the same slice. Center: EYFP expression in the ZI (scale bar: 20 μm). Right: labeled axons in the posterior medial nucleus (POm; bottom; scale bar: 20 μm) and S1 cortex (top; scale bar: 10 μm). (**C**) Quantification of cerebellar nuclei projections to the ZI. Curves show terminal density distribution from dorsal ZI (dZI) to ventral ZI (vZI). Color of curves correspond to images taken along the rostro-caudal axis: green: top (rostral) image; navy: center image; light blue: bottom (caudal) image. Scale bars: 500 μm. (**D**) Labeling strategy for electrophysiological recordings shown in **E–F**. Scale bar: 1 mm. (**E**) Whole-cell patch-clamp recordings from POm-projecting ZI neurons receiving input from cerebellar nuclei. Top: responses to depolarizing current pulses in the absence (left) and presence (right) of TTX (1 μM). Bottom: responses in ZI cells to photostimulation of ChR2-expressing terminals from cerebellar nuclei neurons in the absence (left) and presence (right) of TTX. (**F**) Amplitude (left) and latency (right) of photostimulation-evoked EPSPs before (n=5) and after (n=3) wash-in of TTX. (**G**) Experimental configuration for recordings from mice expressing inhibiting hM4D(Gi) receptors in PV-expressing ZI interneurons. (**H**) In the presence of deschloroclozapine (DCZ), the suppressive effect of climbing fiber (CF) co-activation on acute whisker responses of L2/3 pyramidal neurons is blocked (n=80 cells; N=3 mice). Trace scale bars: 0.1 ΔF/F; 0.5 s; shaded region: SEM across neurons. (**I**) Analysis by neuron. Box: 25th and 75th percentiles; line: median; whiskers: range. Violin plot: kernel density. All summary data, statistical methods, and significance levels are available in *Figure 6—source data 1*.

The online version of this article includes the following source data and figure supplement(s) for figure 6:

**Source data 1.** Summary data, statistical methods, and significance levels for data in *Figure 6*.

**Figure supplement 1.** Cerebellar nuclei do not strongly project to thalamic reticular nucleus.

**Figure supplement 2.** Transsynaptic labeling identifies projections from the cerebellar nuclei to posterior medial nucleus (POm).

**Figure supplement 3.** Whole-cell patch-clamp recordings are performed from neurons in the zona incerta (ZI) that project to posterior medial nucleus (POm) and receive input from cerebellar nuclei.

activation. For this to happen under natural conditions, novel input features have to emerge. Second, the synchronization of CF responses might be critical. Complex spike synchrony is absent or weak during spontaneous activity and might increase with movement initiation (*Welsh, 2002*; *Kitazawa and Wolpert, 2005*) or sensory stimulation (*Schultz et al., 2009*). Synchrony in a sufficiently high number of PCs is needed to evoke inhibition and subsequent rebound excitation in target neurons in the cerebellar nuclei (*Gauck and Jaeger, 2000*; *Person and Raman, 2012*). Parallel fibers (PFs) likely cannot generate synchronous activity in enough PCs due to the spatial arrangement of PF input (*Tang et al., 2019*). A threshold level of synchronization will be reached with optogenetic CF activation but may also naturally be reached with specific qualities of the sensory input signal, e.g., those that make it an error signal. It is also conceivable that synchronization to a threshold level is driven by an error signal that does not originate from whisker stimulation but has another sensory or non-sensory origin.

Whisker-related sensory information in rodents is conveyed by the trigeminal ganglion via the VPM and POm thalamic nuclei to the barrel cortex; information from specific whiskers is projected via VPM to cortical layer 4 (*Feldmeyer et al., 2013*). The observation that cerebellar signals impact S1 cortex via ZI and POm fits the description of a modulatory pathway that does not interfere with thalamic input in layer 4, but with synaptic signal integration in the more superficial as well as deeper layers. Both two-photon imaging and chemogenetics identify SST interneurons in S1 cortex as the ultimate effectors of the activation of this modulatory pathway (note that a similar mechanism might underlie a phenomenon clinically known as CBI or 'cerebellar-brain inhibition,' where electrical or magnetic stimulation over the cerebellum in human subjects causes a reduction in excitability of the motor cortex *Ugawa et al., 1991*; *Ugawa et al., 1995*). SST interneurons, in turn, are regulated by VIP interneurons that receive activation from POm (*Audette et al., 2018*). We identified PV neurons in ZI as neurons that are critically involved in this pathway as well, likely reducing the POm-mediated activation of VIP interneurons in S1 cortex. Our findings demonstrate that cerebellar output may recruit PV neurons in ZI sufficiently well to ultimately cause a removal of VIP interneuron-mediated inhibition of SST interneurons and consequently enhance dendritic inhibition of L2/3 pyramidal neurons. This activation sequence likely requires an initial activation of excitatory neurons in the cerebellar nuclei by output from the cerebellar cortex, which may result from rebound excitation following synchronous PC activity (*Person and Raman, 2012*). Another context where PV neurons in the ZI become activated – via glutamatergic input from S1 cortex – is self-grooming in the orofacial area (*Ge et al., 2024*). It is plausible that under such conditions of predicted sensory input, whisker-related L2/3 pyramidal neuron plasticity is suppressed as well. While our findings identify the cerebellar nuclei to ZI to POm projection as critical to conveying CF signals to S1 cortex, it is likely that more projections that are not tested here can provide links from the cerebellum to S1 cortex. This is suggested by the distinct multi-channel output from the cerebellar nuclei (*Kebschull et al., 2020*). For example, we cannot exclude a participation of direct projections from the cerebellar nuclei to VPM or POm (*Beckinghausen et al., 2023*) (note, however, the prominence of projections that reach the ZI and originate in the cerebellar nuclei, as shown in *Figure 6A–C*; *Figure 6—figure supplement 2*). Moreover, it remains possible that cerebellar output reaches S1 cortex via M1 cortex, as both cortices show strong reciprocal connections (*Mao et al., 2011*) and S1-projecting pyramidal neurons in M1 target VIP interneurons in S1 cortex (*Lee et al., 2013*).

Our data show that activity of cerebellar CFs has an impact on S1 cortex, preventing plasticity of L2/3 pyramidal neurons via the recruitment of local SST interneurons. In our experiments, CFs were artificially (optogenetically) stimulated, and definitive conclusions about CF engagement and its consequences under natural conditions remain open. However, our study demonstrates both the capacity for such an impact on cortical signaling and a pathway that enables this impact. These are requirements for CF signals to act as instructive signals in supervised learning (*Knudsen, 1994*). A condition for such an effect is likely a synchronization of CF responses in groups of PCs (*Gauck and Jaeger, 2000*; *Person and Raman, 2012*), a scenario that provides a qualifying distinction from other CF activity patterns (*Welsh, 2002*; *Kitazawa and Wolpert, 2005*; *Schultz et al., 2009*). The observed external influence enables regulatory control that might safeguard the cerebral cortex from maladaptation, albeit at the cost of exposing it to pathological dysfunction of the olivo-cerebellar system, e.g., sensory over-responsivity that has been described in syndromic autism (*Simmons et al., 2022*).

# Materials and methods

## Key resources table

| Reagent type (species) or resource | Designation | Source or reference | Identifiers | Additional information |
|---|---|---|---|---|
| Strain, strain background (*Mus musculus*) | C57BL/6J | The Jackson Laboratory | Strain #: 000664; RRID:IMSR_JAX:000664 | Common name: B6 |
| Strain, strain background (*Mus musculus*) | B6.129P2-Pvalb<sup>tm1(cre)Arbr</sup>/J | The Jackson Laboratory | Strain #: 017320; RRID:IMSR_JAX:017320 | Common name: B6 PV<sup>cre</sup> |
| Strain, strain background (*Mus musculus*) | STOCK Sst<sup>tm2.1(cre)Zjh</sup>/J | The Jackson Laboratory | Strain #: 013044; RRID:IMSR_JAX:013044 | Common name: Sst-IRES-Cre |
| Strain, strain background (*Mus musculus*) | B6J.Cg-Vip<sup>tm1(cre)Zjh</sup>/AreckJ | The Jackson Laboratory | Strain #: 031628; RRID:IMSR_JAX:031628 | Common name: Vip-IRES-cre (C57BL/6J) |
| Strain, strain background (*Mus musculus*) | B6.Cg-Gt(ROSA)26Sor<sup>tm9(CAG-tdTomato)Hze</sup>/J | The Jackson Laboratory | Strain #: 007909; RRID:IMSR_JAX:007909 | Common name: Ai9 |
| Recombinant DNA reagent | AAV9-CaMKIIa-hChR2(H134R)-mCherry (viral prep) | Addgene | Addgene #: 26975-AAV9; RRID:Addgene_26975 | Titer (GC/ml): $2.4 \times 10^{13}$ |
| Recombinant DNA reagent | AAV9-CaMKIIa-hChR2(H134R)-EYFP (viral prep) | Addgene | Addgene #: 26969-AAV9; RRID:Addgene_26969 | Titer (GC/ml): $3.1 \times 10^{13}$ |
| Recombinant DNA reagent | AAV1.CAG.Flex.GCaMP6f.WPRE.SV40 (viral prep) | Addgene | Addgene #: 100835-AAV1; RRID:Addgene_100835 | Titer (GC/ml): $8.9 \times 10^{12}$ |
| Recombinant DNA reagent | AAV1.Syn.GCaMP6f.WPRE.SV40 (viral prep) | Addgene | Addgene #: 100837-AAV1; RRID:Addgene_100837 | Titer (GC/ml): $7.4 \times 10^{12}$ |
| Recombinant DNA reagent | AAV9-FLEX-tdTomato (viral prep) | Addgene | Addgene #: 28306-AAV9; RRID:Addgene_28306 | Titer (GC/ml): $3.1 \times 1013$ |
| Recombinant DNA reagent | AAV5-hSyn-DIO-hM3D(Gq)-mCherry (viral prep) | Addgene | Addgene #: 44361-AAV5; RRID:Addgene_44361 | Titer (GC/ml): $2.2 \times 10^{13}$ |
| Recombinant DNA reagent | AAV5-hSyn-DIO-hM4D(Gi)-mCherry (viral prep) | Addgene | Addgene #: 44362-AAV5; RRID:Addgene_44362 | Titer (GC/ml): $2.3 \times 10^{13}$ |
| Recombinant DNA reagent | AAV1-EF1a-Flpo (viral prep) | Addgene | Addgene #: 55637-AAV1; RRID:Addgene_55637 | Titer (GC/ml): $8.9 \times 10^{12}$ |
| Recombinant DNA reagent | hSyn-Coff/Fon hChR2(H134R)-EYFP (plasmid) | Addgene | Plasmid #: 55648; RRID:Addgene_55648 | Packaged as AAV1 by UNC Vector Core. Titer (GC/ml): $2.4 \times 10^{13}$ |
| Recombinant DNA reagent | AAV5-hSyn-hChR2(H134R)-EYFP (viral prep) | Addgene | Addgene #: 26973-AAV5; RRID:Addgene_26973 | Packaged as AAV5 by UNC Vector Core. Titer (GC/ml): $5.3 \times 10^{12}$ |
| Recombinant DNA reagent | pAAV-sL7-Cre-HA-WPRE (plasmid) | Addgene | Plasmid #: 204488; RRID:Addgene_204488 | Packaged as AAV1 by Princeton Neuroscience Institute Viral Core Facility. Titer (GC/ml): $3.0 \times 10^{13}$ |
| Chemical compound, drug | Fluoro-Ruby | Invitrogen | Thermo Fisher Scientific catalog #: D1817 | |
| Chemical compound, drug | Fluoro-Gold | Fluorochrome | Fluorochrome cat # Fluoro-Gold; US Patent No. 4, 716,905 | |
| Antibody | anti-Calbindin (Guinea pig polyclonal) | Synaptic Systems GmBH | Synaptic Systems GmBH Catalog #: 214 004; RRID:AB_10550535 | IHC: 1:500 |
| Antibody | anti-Guinea Pig IgG (Donkey polyclonal) | Jackson ImmunoResearch Labs | Jackson ImmunoResearch Labs Catalog #: 706-165-148; RRID:AB_2340460 | IHC: 1:200 Conjugate: Cyanine Cy3 |

*Continued on next page*

*Continued*

| Reagent type (species) or resource | Designation | Source or reference | Identifiers | Additional information |
|---|---|---|---|---|
| Antibody | anti-VGLUT2 (Rabbit polyclonal) | Thermo Fisher Scientific | Thermo Fisher Scientific Cat# 42–7800, RRID:AB_2533537 | IHC: 1:500 |
| Antibody | anti-Rabbit IgG (Donkey polyclonal) | Jackson ImmunoResearch Labs | Jackson ImmunoResearch Labs Cat# 711-605-152, RRID:AB_2492288 | IHC: 1:500 Conjugate: Alexa Fluor 647 |
| Software, algorithm | Fiji | PMID:22743772 | RRID:SCR_002285 | Version: 2.9.0/1.53t |
| Software, algorithm | MATLAB | PMID:30609523, PMID:21934110 | RRID:SCR_001622 | Version R2017b (data collection and preprocessing) and version R2022a (data analysis) |
| Software, algorithm | GraphPad Prism | GraphPad | RRID:SCR_002798 | Version: 7.0 |

## Animals

All animal experiments were approved and conducted in accordance with the regulations and guidelines of the Institutional Animal Care and Use Committee of the University of Chicago (IACUC 72496). Mice were housed on a 12 hr light/dark cycle and fed a standard rodent diet. Animals of either sex were used in all experiments and no sex-dependent differences were observed in any reported measures. Strains included wildtype (B6) mice (C57BL/6J; Jax: 000664; The Jackson Laboratory, Bar Harbor, ME), B6 PV$^{cre}$ mice (B6.129P2-Pvalbtm1(cre)Arbr/J; Jax: 017320; The Jackson Laboratory), Sst-IRES-Cre mice (STOCK Ssttm2.1(cre)Zjh/J; Jax: 013044; The Jackson Laboratory), Vip-IRES-cre mice (B6J.Cg-Viptm1(cre)Zjh/AreckJ; Jax: 031628; The Jackson Laboratory), and Ai9 mice (B6. Cg-Gt(ROSA)26Sortm9(CAG-tdTomato)Hze/J; Jax: 007909; The Jackson Laboratory). Homozygous Sst-IRES-Cre, B6 PV$^{cre}$, and Vip-IRES-Cre animals were bred with wild-type animals to generate heterozygotes. To generate SST and VIP reporter lines used in a subset of experiments (Sst-IRES-Cre$^{tdTomato}$ and Vip-IRES-Cre$^{tdTomato}$), homozygous Sst-IRES-Cre or Vip-IRES-Cre mice were bred with homozygous Ai9 mice. Only animals heterozygous for Sst-IRES-Cre or Vip-IRES-Cre were used across all experiments given evidence that endogenous somatostatin (SST) and vasoactive intestinal polypeptide (VIP) expression may be reduced in homozygous Sst-IRES-Cre and Vip-IRES-Cre animals, respectively (*Viollet et al., 2017*; *Cheng et al., 2019*).

## Stereotaxic injections

A list of all viruses with titers is provided in the Key resources table above.

### For electrophysiological experiments

Stereotaxic virus injections for electrophysiological experiments were performed as previously described using C57BL/6J mice between postnatal day 23–28 (*Koster and Sherman, 2024*). For optogenetics experiments in which cerebellar inputs to zona incerta were measured, a 250 nL (40 nL/min) infusion of AAV5-hSyn-hChR2(H134R)-EYFP (Addgene; packaged by UNC Vector Core, Chapel Hill, NC) was made into the right deep cerebellar nuclei (–1.5 ML; –5.8 AP; –3.2 DV *Paxinos and Franklin, 2019*) using a 1 mL Hamilton syringe (catalog: 65458–02) with a flat needle tip. Next, a small infusion (40 nL total, at 7 nL/min) of fluoro-Ruby (Thermo Fisher Scientific, Waltham, MA; catalog #: D1817) was made into left posterior medial thalamus using a 0.5 mL Hamilton syringe (Hamilton Company, Reno, NV; catalog: 65457–01). Following a two-week incubation, animals were sacrificed for recordings in acute slices (see *Slice electrophysiology*).

### For tracing experiments

For anterograde transsynaptic labeling experiments, a dual-injection approach was utilized to specifically label the outputs of neurons in the zona incerta that receive input from the contralateral cerebellar nuclei. Flpo-recombinase (AAV1-EF1a-Flpo, item#: 55637, Addgene, Watertown, MA) was injected into left deep cerebellar nuclei as above (+1.5 ML; –5.8 AP; –3.2 DV), followed by an infusion of AAV1-Flp-on ChR2-EYFP (hSyn-Coff/Fon hChR2(H134R)-EYFP, Addgene; packaged by UNC Vector Core) into right zona incerta (–1.8 ML; –2.0 AP; –4.0 DV). Following a 5 week incubation period – the

period required for anterograde transport of the AAV1 virus injected into zona incerta – animals were transcardially perfused for histology (see *Tissue preparation for fluorescence microscopy and immunohistochemistry*).

To visualize potential overlap between cerebellar terminals and cells in thalamic nuclei (i.e. the thalamic reticular nucleus) that project to somatosensory thalamus (ventral posteromedial nucleus and posterior medial thalamus), an orthograde virus (AAV5-hSyn-hChR2(H134R)EYFP, Addgene) was injected into the right deep cerebellar nuclei (–1.5 ML; –5.8 AP; 3.2 DV) and fluoro-Gold (2% w/v in sterile saline, Fluorochrome LLC, Denver, CO) was injected into the left somatosensory thalamus. Following a 2 week incubation, animals were transcardially perfused for histology (see *Tissue preparation for fluorescence microscopy and immunohistochemistry*).

## For two-photon calcium imaging experiments

Viral injections were performed to express channelrhodopsin-2 in cerebellar climbing fibers; fluorescent calcium indicators in cerebellar Purkinje cells or primary somatosensory (S1; barrel) cortex neurons; tdTomato, activating DREADDs, or inactivating DREADDs in S1 cortex; and inactivating DREADDs in interneurons in zona incerta. With respect to the inferior olive injection site (conferring viral expression in climbing fibers terminating in the contralateral cerebellar hemisphere), injections into cerebellar cortex were performed contralaterally, and injections into zona incerta and somatosensory cortex were performed ipsilaterally.

To express channelrhodopsin-2 in cerebellar climbing fibers, ChR2-EYFP (AAV9-CaMKIIahChR2 (H134R)-EYFP; Addgene #26969-AAV9) or ChR2-mCherry (pAAV9-CaMKIIahChR2(H134R)-mCherry; Addgene #26975-AAV9) was injected into the left inferior olive as follows: after surgical preparation (see *Two-photon calcium imaging: Cranial window surgeries*), tissues attached to the occipital bone were surgically detached, followed by opening of the posterior atlanto-occipital membrane to expose the dorsal medulla. A 5 µL Hamilton Neuros syringe (catalog: 65460–03) with a 33-gauge beveled needle tip (catalog: 65461–02) was inserted into the left medulla at an angle of 58–59° to the perpendicular line (*Figure 1—figure supplement 1A*), 0.75 mm laterally and 1.70 mm deep from the surface of the medulla as previously described (*Miyazaki and Watanabe, 2011*). A total volume of 400 nL was infused at a rate of 80 nL/min, and the syringe was held in place for 5 min before removal. ChR2-EYFP or ChR2-mCherry was chosen based on the excitation and emission spectra of other fluorophores expressed in the same animal to maximize signal separation when conducting post-hoc confocal imaging.

Stereotaxic virus injections for imaging of Purkinje cell dendrites were administered as previously described using P80-120 adult C57BL/6 J mice (*Lin et al., 2024*). A virus mixture was prepared with 0.5% L7-Cre (pAAV-sL7-Cre-HA-WPRE, Addgene plasmid #:204488, packaged as AAV1 by the Princeton Neuroscience Institute Viral Core Facility, Princeton, NJ), and 20% Cre-dependent GCaMP6f (AAV1.CAG.Flex.GCaMP6f.WPRE.SV40; Addgene, #100835-AAV1) in saline solution. A glass pipette with a tip diameter of ~300 µm was prepared by a puller (model P-97, Sutter Instrument, Novato, CA) and used to inject a total volume of 1800 nL into two separate injection sites: 900 nL were injected into right medial Crus I (–1.8 ML; –7 AP; –1.8 DV), followed by a 900 nL injection into right medial Crus II (–2.5 ML; –7 AP; –2.2 DV). The pipette carrying the AAV was held in place for 5 min before withdrawal.

For DREADD suppression of PV interneurons in zona incerta, a 1 mL Hamilton syringe was used to infuse Cre-dependent hm4D(Gi) (AAV5-hSyn-DIO-hM4D(Gi)-mCherry; Addgene #44362-AAV5) into the left zona incerta (1.8 ML; –2.0 AP; –4.0 DV) of B6 PV^cre animals.

To express GCaMP6f in primary somatosensory (barrel) cortex neurons in C57BL/6 J, Sst-IRES-Cre^tdTomato, and Vip-IRES-Cre^tdTomato animals, a 5 µL Hamilton Neuros syringe with a 33-gauge beveled needle tip was used to infuse a total volume of 800 nL of GCaMP6f (AAV1.Syn.GCaMP6f.WPRE. SV40; Addgene #100837-AAV1) at two separate injection sites: 400 nL were injected at –1.2 AP; 2.5 ML, and 400 nL at –1.5 AP; 3 ML. At each injection site, 200 nL were first injected 300 µm below the pial surface, followed by 200 nL at 150 µm below the pial surface. The injections were performed at a rate of 100 nL/min, and the syringe was held in place for 5 min before removal. To express GCaMP6f in barrel cortex neurons with co-expression of tdTomato in parvalbumin (PV)-, SST-, and VIP-expressing interneurons in B6 PV^cre, Sst-IRES-Cre, or Vip-IRES-Cre animals, respectively, GCaMP6f (Addgene #100837-AAV1) and Cre-dependent tdTomato (AAV9-FLEX-tdTomato; Addgene #28306-AAV9) were co-injected using the same technique. To optimize co-expression of the viruses, GCaMP6f

with tdTomato was mixed at a 3:1 volume ratio. For DREADD activation or suppression of VIP, SST, or PV interneurons, Cre-dependent hM3D(Gq) (AAV5-hSyn-DIO-hM3D(Gq)-mCherry; Addgene #44361-AAV5) or Cre-dependent hM4D(Gi) (AAV5-hSyn-DIO-hM4D(Gi)-mCherry; Addgene #44362-AAV5) were used in place of tdTomato, respectively.

For all two-photon imaging experiments, recordings began after a 4 week incubation period after the final virus injection (see *Two-photon calcium imaging: cranial window surgeries for detail*).

## Tissue preparation for fluorescence microscopy and immunohistochemistry

After completion of two-photon experiments, selected animals (all those used in DREADD and cerebellar imaging experiments, and a subset of animals in all other experimental cohorts) were sacrificed for post-hoc characterization of viral expression. Animals were transcardially perfused with ice-cold 4% paraformaldehyde in phosphate-buffered saline (25 mL), pH 7.4. The brain was extracted and postfixed in 4% paraformaldehyde overnight at 4°C before being transferred to a cold 30% sucrose (in phosphate-buffered saline) solution for >48 hr. Brains were then cryosectioned at 50 µm thickness, with an approximately equal number of brains sectioned coronally or sagittally per experimental cohort. For tracing experiments, animals were perfused with ice-cold phosphate-buffered saline (25 mL) prior to perfusion with paraformaldehyde and cryosectioned coronally at 150 µm thickness. Slices were stored in phosphate-buffered saline prior to immunohistochemistry or confocal imaging.

For post-hoc visualization of Purkinje cells in mice expressing ChR2-EYFP in climbing fibers (i.e. in animals without GCaMP6f expression in Purkinje cells), calbindin immunohistochemical staining was performed. Tissue slices were washed in a 0.01 M phosphate-buffered saline solution containing 200 mM glycine for 2 hr at 4°C, then incubated in a 0.01 M phosphate-buffered saline solution containing 10 mM sodium citrate and 0.1% Tween-20 at ~60°C for 30 min using a heated water bath. After cooling to room temperature, the slices were washed three times in a 0.01 M phosphate-buffered saline solution containing 0.5% Tween-20 (PBS-Tween) at room temperature for a period of 15 min per wash. Tissue was permeabilized with 200 mM glycine in PBS-Tween for 15 min at room temperature. Blocking was done for 2 hr at room temperature with a PBS-Tween solution containing 5% bovine serum albumin and 10% normal donkey serum, followed by overnight incubation (12–15 hr) at 4°C in a PBS-Tween solution containing 1% normal donkey serum and anti-Calbindin (Guinea pig polyclonal) primary antibody (1:500; Synaptic Systems Cat# 214 004, RRID:AB_10550535; Synaptic Systems GmbH, Germany). After washing the tissue three times for 15 min in PBS-Tween at 4°C, slices were incubated for 2 hr at 4°C in PBS-Tween containing 1% normal donkey serum and anti-Guinea Pig IgG (Donkey polyclonal) conjugated with Cyanine Cy3 secondary antibody (1:200; Jackson ImmunoResearch Labs Cat# 706-165-148, RRID:AB_2340460; Jackson ImmunoResearch Labs, West Grove, PA). Finally, the slices were washed three times in PBS-Tween (10 min per wash) before mounting with Vectashield (Vector Laboratories, Inc). The mounted slices were allowed to set overnight before imaging.

To confirm the channelrhodopsin-2 expression observed in cerebellar cortex was indeed expressed in climbing fiber terminals, a VGluT2 was additionally labeled in a subset of tissue. Anti-VGLUT2 (Rabbit polyclonal) primary antibody (1:500; Thermo Fisher Scientific Cat# 42–7800, RRID:AB_2533537) was added to the primary antibody solution above and anti-Rabbit IgG (Donkey polyclonal) conjugated with Alexa Fluor 647 (1:500; Jackson ImmunoResearch Labs Cat# 711-605-152, RRID:AB_2492288) was added to the secondary antibody solution above.

## Confocal imaging

### Image acquisition

Z-stack confocal images were taken at 5x (Olympus MPlan N 0.1NA, air; Olympus Corporation, Japan), 10x (Zeiss Achroplan 0.25NA, air; Carl Zeiss AG, Germany), 20x (Olympus UMPlanFL N 0.5NA, water), 40x (Zeiss EC Plan-Neofluar 1.3NA, oil immersion), and 63x (Zeiss W Plan-Apochromat, water) magnifications with a Zeiss LSM 900 Axio Examiner.Z1 scope. A subset of images (*Figure 1E*, *Figure 1—figure supplement 1B*) were taken with a Zeiss LSM 5 Exciter, Axioskop 2 (using the 20x Olympus UMPlanFL and 40x Zeiss EC Plan-Neofluar objectives, respectively). The images in *Figure 6D* (shown with channel separation in *Figure 6—figure supplement 3*) and *Figure 6—figure supplement 2* were taken with a second Zeiss LSM 900 confocal microscope. To facilitate direct comparison, images

shown in series (as in *Figure 6C*, *Figure 6—figure supplement 2D and E*) were taken with identical hardware and software configurations, available in the image metadata.

## Post-hoc image processing

To stitch multiple images (as in *Figure 1B and D*, *Figure 2D–F* (right), *Figure 4B and F*, *Figure 6B and C*, (top), *Figure 1—figure supplement 1E*, *Figure 6—figure supplement 1A*, *Figure 6—figure supplement 1B* (top and bottom left), and *Figure 6—figure supplement 2C–E*), a series of pairwise stitches were performed using Fiji (RRID:SCR_002285). *Figure 1B*, *Figure 1—figure supplement 1A* (top) each consisted of four images (each 5x magnification, 1024×1024 pixels). *Figure 1D* consisted of 12 images (each 5x magnification, 512×512 pixels). *Figure 2D* (right) consisted of 10 images (each 10x magnification, 1024×1024 pixels). *Figure 2E and F* (right) each consisted of eight images (each 10x magnification, 512×512 pixels). *Figure 4B and F* each consisted of two images (each 20x magnification, 512×512 pixels). *Figure 6B* (also shown as the first image in *Figure 6—figure supplement 3D*), and all images in *Figure 6—figure supplement 2D and E* each consisted of six images (each 5x magnification, 512×512 pixels). Images in *Figure 6C* each consisted of nine images (each 10x magnification, 512×512 pixels). *Figure 1—figure supplement 1E* consisted of eight images (each 5x magnification, 1024×1024 pixels). *Figure 6—figure supplement 2C* consisted of 10 images (each 40x magnification, 512×512 pixels). Note that a subset of the final stitched images displayed in figures are cropped to show relevant detail.

Channel colors set during image acquisition followed conventional emission wavelength assignments (i.e. red to indicate tdTomato, green to indicate GCaMP6f, etc.). To illustrate interneuron subtypes and continuity across experiments (e.g. cyan to indicate ChR2 expression across all optogenetic stimulation experiments), channel colors were specified post-imaging using Fiji by separating channels and selecting the appropriate color when merging, ignoring source LUTs. When necessary, post-hoc brightness and contrast adjustments were applied linearly to an entire image – *after* stitching, if applicable – using Fiji. To facilitate direct comparison, identical linear brightness/contrast adjustments were made across all images shown in series (as in *Figure 6C*, *Figure 6—figure supplement 2D*, and *Figure 6—figure supplement 2E*).

To measure terminal density across the sections shown in *Figure 6C*, Fiji was used to rotate each image such that the axis separating ventral from dorsal zona incerta was horizontal. A 400 µm × 1 mm region of interest (ROI) was drawn around the zona incerta, and each row of pixels within this ROI was separated to facilitate calculation of the integrated density across each row of pixels. A similar method was used to quantify the terminal density of cerebellar inputs to zona incerta as in *Figure 6—figure supplement 3*. Briefly, images of sections matched for their rostro-caudal location were rotated as before. Then, five line-ROIs oriented vertically, each being 200 µm wide, were positioned side-by-side across the mediolateral axis of the zona incerta, and the spatial pattern of fluorescence along these lines was measured by calculating the mean gray value along their length (which was roughly 400 µm from the dorsal boundary to the ventral boundary). These were averaged for each animal used. Shading represents SEM for both measurements.

## Slice electrophysiology

### Acute slice preparation and whole cell recordings

For recordings in acute slice, injected animals (see *Stereotaxic injections: For electrophysiological experiments*) were deeply anesthetized using the isoflurane drop method in a bell jar (with a raised platform for the animal) and immediately transcardially perfused with 8–10 mL of ice-cold oxygenated (95% $O_2$, 5% $CO_2$) artificial cerebrospinal fluid, which contained the following (in mM): 125 NaCl, 25 $NaHCO_3$, 3 KCl, 1.25 $NaH_2PO_4$, 1 $MgCl_2$, 2 $CaCl_2$, and 25 glucose. The brain was then extracted, glue-mounted on a vibratome (Leica Biosystems, Germany) platform and blocked for coronal slices using an agarose (5%) cube and sliced in the same solution (ice-cold). Slices were cut to 365 µm thickness. The brain slices were then transferred to 32–34°C oxygenated artificial cerebrospinal fluid that was allowed to return to room temperature over the course of 1 hr, which constituted the slice recovery period.

Slices containing the zona incerta and terminals from the cerebellar nuclei inputs were visualized using differential interference contrast with a Zeiss Axioskop 2FS microscope. Fluorescence from ChR2-EYFP expression and from retrograde fluoro-Ruby labeling was confirmed using the 5x air

objective and guided recording locations in zona incerta. Recordings were made with a Multiclamp 700B amplifier and pCLAMP software (Molecular Devices, San Jose, CA). Recording glass pipettes with 7–9 MΩ resistance were filled with an internal solution as follows (in mM): 117 K-gluconate, 13 KCl, 1 MgCl$_2$, 0.07 CaCl$_2$, 10 Hepes, 0.1 EGTA, 2 Na$_2$-ATP, 0.4 Na-GTP, pH 7.3, 290 mOsm. Incertal cells whose morphology was clearly revealed by strong red fluorescence, indicating retrograde labeling from posterior medial thalamus, were targeted for recordings. Bath application of TTX (1 μM) was performed in a subset of recorded cells to verify the monosynaptic nature of the cerebellar input to zona incerta.

Optogenetic stimulation was delivered using a 355 nm laser (DPSS: 3505–100), controlled with galvanometer mirrors (Novanta, Bedford, MA) focused on the slice through a 5x air objective using custom software in MATLAB (The MathWorks, Inc, Natick, MA; RRID:SCR_001622). Focal photostimulation of the ChR2-expressing synaptic terminals in zona incerta was performed at 10 Hz (four pulses of 1 ms duration at 100 ms interstimulus interval).

## Electrophysiological data analysis

Electrophysiological data were collected using custom MATLAB software and analyzed using Graphpad Prism (GraphPad by Dotmatics, Boston, MA; version 7.0; RRID:SCR_002798). The amplitude of excitatory responses (i.e. the first response in the stimulus train) to stimulation pulses was measured by subtracting the average value for 20 ms before the delivery of a pulse (baseline) from the maximum value of the peak in current clamp at resting membrane potential. Latency of the optogenetic response was calculated by measuring the delay in milliseconds from the onset of the first optogenetic stimulus to the moment the resulting response reached >10% of its maximum amplitude.

## Two-photon calcium imaging

### Cranial window surgeries for imaging of Purkinje cell dendrites

To minimize overall recovery burden, surgeries were performed in two stages: (1) inferior olive virus injection and headframe installation, and (2) cerebellar virus injection and cerebellar cranial window installation as previously described (*Lin et al., 2024*). In the first stage, mice (P60-90) were deeply anesthetized using 1–2% isoflurane and clamped by ear bars at the external acoustic foramen. The line between the clamping point and maxilla was set parallel to the horizontal plane. After trimming the fur on top of the skull with clippers, the surgical site was prepared by applying betadine and 70% ethanol three times in an alternating fashion. Meloxicam (2 mg/kg), extended-release buprenorphine (0.1 mg/kg), and 0.5 mL saline were administered subcutaneously, and depth of anesthesia was confirmed via tactile stimulation of the toe before making a 20–25 mm incision in the skin to reveal the occipital bone and attached tissues. After opening of the posterior atlantooccipital membrane and channelrhodopsin-2 virus injection into the left inferior olive (see *Stereotaxic injections: For two-photon calcium imaging experiments*), tissues were reattached to the occipital bone using instant adhesive, ensuring adequate space for later implantation of the cranial window over the right cerebellar cortex. A custom titanium headframe (H. E. Parmer Company, Nashville, TN) was secured with dental cement (Stoelting Co., Wood Dale, IL), completing the first stage.

After monitoring recovery for a period of 4 days, mice underwent the second surgery. A circular craniotomy over right cerebellar cortex with a diameter of 4 mm, centered at –2.7 ML; –6.9 AP, was performed using a dental drill. The dura was carefully removed to expose lobules Crus I and anterior Crus II before administering GCaMP6f injections (see *Stereotaxic injections: For two-photon calcium imaging experiments*). Following injection, a two-layer glass window was installed using C&B Metabond dental cement (Patterson Dental Company, Saint Paul, MN), completing the second stage. The glass windows consisted of a 4 mm glass window (Tower Optical Corp, Boynton Beach, FL, # 4540–0495) adhered to a 5 mm glass window (Warner Instruments, Holliston, MA # CS-5R) using ultraviolet light-activated glue (Norland Optical Adhesive 71; Norland Products Inc, Jamesburg, NJ). Experiments were conducted 4.5 weeks after inferior olive injection and 4 weeks after GCaMP6f injection.

### Cranial window surgeries for imaging of barrel cortex neurons

Surgeries were again separated into two stages: (1) inferior olive virus injection, headframe installation, and cerebellar cranial window installation; and (2) S1 cranial window installation and virus injection. First, channelrhodopsin-2 was injected into the left inferior olive as above (see *Cranial window*

*surgeries for imaging of Purkinje cell dendrites* and *Stereotaxic injections: For two-photon calcium imaging*). After reattaching tissues to the occipital bone using instant adhesive, a dental drill was used to perform a circular craniotomy over right cerebellar cortex (lobules Crus I and II), followed by removal of the dura and installation of the glass window as above. Finally, the custom titanium head-frame was secured using dental cement, completing the first stage.

After monitoring recovery for a period of 4 days, mice underwent the second surgery. After surgically detaching a small portion of the left temporalis muscle attached to the parietal bone, a second circular craniotomy over left primary somatosensory cortex with a diameter of 4 mm (centered at –1.5 AP; 3 ML) was performed. The dura was removed to expose the neocortical surface before administering the appropriate viral injections (see *Stereotaxic injections: For two-photon calcium imaging experiments*). Following injection, a glass window was installed over the craniotomy using dental cement. To ensure optical isolation during recordings, a custom sheath was used to couple an optical patch cable (used for LED stimulation of cerebellar climbing fibers; see *Optogenetic and tactile stimulation*) to the cerebellar window. This sheath was adhered over the cerebellar window using Metabond dental cement, completing the second stage. For experiments involving DREADD manipulation of neurons in zona incerta, viral injections into left zona incerta were administered three days prior to the first surgery. Thus, experiments were conducted approximately 5 weeks after zona incerta injection, 4.5 weeks after inferior olive injection, and 4 weeks after primary somatosensory cortex injections.

## Habituation

After completion of surgeries, postoperative care was provided for 5–7 days. Once animals exhibited exploratory behavior without signs of pain or distress, habituation began with handling of mice for 15 mins and transportation from the animal facility to the laboratory. After two days, anxiety during handling typically diminished, and the handling period increased to 30 min in the vicinity of the recording area. Following two days of handling, the mice were introduced to a free-running treadmill and head restraint via clamping of the head frame on left and right sides. The duration of time spent on the treadmill increased from 10 min to 2 hr per day depending on the mouse's comfort. The relative time spent running on the treadmill typically decreased after two days. To habituate mice to air puff stimulation during this time, receptive field mapping of the field of view was conducted by delivering 8 psi air puffs through a capillary tube while observing epifluorescence signals and calcium responses in real time. Air puffs were delivered at irregular frequencies, with at least 30 s between stimuli, to avoid influencing subsequent plasticity experiments. In total, mice were habituated for a period of 1–2 weeks until they showed no aversive responses to air puff stimulation, spent the majority of time in the recording environment at rest, and reduced movement initiation (whisking/running) with air puff stimulation. Animals used for DREADD manipulation experiments were familiarized with the DCZ injection procedure by receiving DMSO/saline injections during habituation for two days prior to recording.

## Optogenetic and tactile stimulation

To optogenetically stimulate cerebellar climbing fibers, the tip of a Ø200 μm core, 0.39 NA rotary joint patch cable (Thorlabs catalog: RJPSL2; Thorlabs, Inc, Newton, NJ) was inserted into the sheath attached to the cranial window over the right cerebellar cortex (see *Two-photon calcium imaging: Cranial window surgeries*). For experiments in which Purkinje cell responses to optogenetic climbing fiber stimulation were also measured, the patch cable was inserted into a custom light shield that fit over both the objective and the cable. The tip of the patch cable was centered over Crus II in all optogenetic stimulation experiments (–2.7 ML; –6.9 AP). Blue light pulses were delivered with a 470 nm Fiber-Coupled LED (Thorlabs catalog: M470F3). The absolute power output was adjusted to 2.5 mW. Given the tip of the LED was positioned 1 mm over the surface of the brain (0.7 mm over the surface of the optical window), the estimated spot size was 0.6 mm and the estimated power density was 8 mW/mm$^2$ at the pial surface. To deliver whisker stimuli, a 0.86 mm diameter glass capillary tube was positioned over the right C-row whiskers 5 mm from the right whisker pad at an angle of 45° in the X and Y planes (adjusted slightly during receptive field mapping for each animal). Air puffs were delivered at 8 psi (Picospritzer III, Parker Hannifin, Cleveland, OH). For all imaging sessions, calcium activity was measured in trials lasting 20 s, with a stimulus delivered once per trial. Trials began at random intervals, with an interval of at least 5 s between trials. When testing Purkinje cell responses

to optogenetic climbing fiber stimulation, 470 nm pulses were delivered with the following durations: 1×20 ms, 1×50 ms, 3×15 ms at 8 Hz, and (3×15 ms at 8 Hz)*2 with a 250 ms interval between epochs. The 1×50 ms light pulse was selected for all subsequent experiments given this stimulation duration evoked responses resembling spontaneous events in the same Purkinje cells (*Figure 1—figure supplement 1*). To test the basic transmission of the climbing fiber signal to S1 (i.e. the immediate influence of optogenetic stimulation of climbing fibers on acute responses to whisker stimulation shown in *Figure 2*), 100 ms air puff test pulses were delivered with or without optogenetic co-stimulation of climbing fibers. For paired test pulse stimuli, the 50 ms blue light stimulus was delivered with a delay of 45 ms with respect to the onset of the whisker stimulus. This delay was chosen to mimic the natural latency of climbing fiber responses to whisker stimulation, which peaks approximately 50 ms after stimulus onset (*Bosman et al., 2010*). For plasticity experiments, whisker test pulses were delivered during a baseline period of at least 40 min, followed by a 5 min period of rhythmic whisker stimulation (RWS) at 8 Hz – the speed at which mice naturally sample objects (*Diamond et al., 2008*). For plasticity experiments with optogenetic co-activation of climbing fibers (RWS+CF), the same protocol was followed with the addition of 50 ms blue light pulses at 1 Hz during plasticity induction. The 8 Hz whisker stimulation and 1 Hz CF stimulation occurred at a consistent time relative to one another, with a blue light pulse always occurring at a 45 ms delay relative to the onset of a whisker stimulus (the same delay used for the paired test pulses). After RWS or RWS+CF, whisker test pulses were delivered for a period of at least 60 min. To coordinate precise stimulus delivery, a Cygnus Digital Stimulator (catalog: PG4000A, Cygnus Technology, Inc, Delaware Water Gap, PA) was interfaced with an Arduino Uno (Qualcomm, San Diego, CA) to trigger both the LED driver and picospritzer.

## Cerebellar imaging protocol

Calcium imaging of the genetically encoded indicator GCaMP6f was conducted in Crus I/II of the right hemisphere of awake, head-fixed mice using a laser scanning two-photon microscope (Neurolabware, Los Angeles, CA, USA) and Scanbox software (Scanbox, Los Angeles, CA). Calcium images were obtained at a frame rate of 30.98 fps with a pixel dimension of 512 × 796, using an 8 KHz resonant scanning mirror with bidirectional scanning. A Mai Tai DeepSee (Spectra-Physics, Milpitas, CA) laser source was used to excite GCaMP6f at 920 nm. The fluorescence emission was collected through a 16x water immersion objective (Nikon LWD 0.8NA, 3 mm WD) using a GaAsP PMT (Hamamatsu Photonics, Shizuoka, Japan). A 2.0 or 2.8x digital zoom was applied during imaging, generating a field of view of 760 × 613 or 551 × 444 µm, respectively. To minimize background noise originating from ambient light, a custom light shield was fitted around the brain window and objective. This shield had a small opening such that the Thorlabs patch cable used for LED stimulation of climbing fibers could be positioned over the cerebellar window. The laser power was set to 1% with a PMT gain between 0.79 and 0.81, allowing for prolonged recording while minimizing phototoxicity.

## S1 imaging protocol

Calcium imaging of GCaMP6f was conducted in barrel cortex of the left hemisphere using the same hardware as above. Images were obtained at a frame rate of 30.98 Hz with a pixel dimension of 512 × 796. A 2.4x digital zoom was applied during imaging, generating a field of view of 638 × 515 µm. Two of 11 RWS+CF experiments and one of 10 RWS experiments were conducted using a 2.0x digital zoom (generating a field of view of 760 × 613 µm) and were added to the experimental cohorts, as results were not significantly different across any reported measures. A custom light shield was fitted around the brain window and objective to reduce background noise originating from ambient light. Note further isolation of the LED light used for optogenetic stimulation of climbing fibers in these recordings was provided by the sheath attached to the cerebellar window (see *Cranial window surgeries for imaging of barrel cortex neurons*). While observing the epifluorescence signal, small vessels on the putative surface of the brain were put into focus in the z plane and matched to a field of view selected during receptive field mapping (see *Habituation*). After adjusting the focus 250 µm below the putative pial surface, small adjustments were made in the x-y planes while observing calcium signal in real time to maximize responsivity to whisker stimulation and overlap between GCaMP6f and tdTomato or mCherry in the field of view. GCaMP6f was excited at 920 nm and tdTomato or mCherry (for DREADD experiments) at 1040 nm, with emitted fluorescence collected using two GaAsP PMTs. The laser power was set to 2%, and adjustments to the PMT gain were made due to slight variance

in the level of GCaMP6f and tdTomato/mCherry signal across animals. PMT gain in the green channel was 0.73 (SD: 0.07), while PMT gain in the red channel was 0.83 (SD: 0.14), which were not significantly different across experimental groups. Responsivity to whisker stimulation was determined by intermittently delivering 8 psi air puffs at random intervals, with a minimum of 30 s between stimuli, to avoid influencing plasticity experiments. If more than 10 min were spent determining an optimal field of view on the day of recording, the experiment was aborted and instead conducted the following day. Mice were occasionally used for multiple recording configurations but were sacrificed immediately after performing plasticity experiments, and selected animals were used for histological characterization (see *Tissue preparation for fluorescence microscopy and immunohistochemistry*). Note DREADD expression was histologically verified in all animals to ensure expression was restricted to neocortex.

## DREADD experimental protocol

To activate the hM4D(Gi) and hM3D(Gq) receptors, we used deschloroclozapine dihydrochloride (DCZ, MedChemExpress, Monmouth Junction, NJ), which was chosen over clozapine N-oxide (CNO) as an agonist due to its heightened selectivity for hM4D(Gi) and hM3D(Gq) receptors over endogenous receptors, a more rapid onset than CNO, and significantly increased potency (**Nagai et al., 2020**). DCZ was dissolved in DMSO at a 0.02 mg/mL concentration and stored at −80°C. On the day of recording, DCZ aliquots were thawed to room temperature and diluted to 0.01 mg/mL with DMSO/ saline. Two-photon calcium imaging experiments began with recordings of baseline responses to whisker stimulation for at least 40 min (with and without optogenetic stimulation of climbing fibers in a subset of experiments), followed by an injection of DCZ (0.1 mg/kg) intraperitoneally. For DREADD inhibition of PV neurons, a dose of 0.02 mg/kg was used instead, as the 0.1 mg/kg dose caused significant increases in neocortical excitability (note PV animals that had received a dose of 0.1 mg/kg were not used for any subsequent experiments or in any of the datasets shown here). Injections were carefully administered to ensure the field of view would not be disturbed; experiments were aborted if this occurred, as responses in each period could no longer be directly compared. Activation of receptors was allowed to take effect for 15 min prior to resumption of recording, after which a second period of baseline activity was recorded. This period lasted between 15 and 30 min, such that a period of fifteen minutes of stable activity was recorded prior to plasticity induction. Plasticity induction protocols were followed as above (see *Optogenetic and tactile stimulation*), and whisker test pulses were delivered for at least 40 min after plasticity induction. To ensure observed plasticity phenomena were not the result of prolonged changes in activity related to hM4D(Gi) or hM4D(Gq) activation, control experiments lasting the same duration as the plasticity experiments were conducted similarly, except without induction of plasticity (see *Figure 4—figure supplement 1*). To ensure DREADD expression was localized to the neocortex, viral spread was histologically characterized in all animals (see *Tissue preparation for fluorescence microscopy and immunohistochemistry*).

## Characterization of neurons

Calcium images taken in the green (PMT0) and red (PMT1) channels were first merged, trial-averaged, and concatenated using Fiji. The average image was then calculated, followed by identification of all cells expressing tdTomato or mCherry (applicable to all recordings except the zona incerta manipulation experiments). ROIs were manually drawn around each cell co-expressing GCaMP6f and tdTomato or mCherry. All images across a session were then concatenated and trial-averaged before calculating the maximum image. ROIs were manually drawn around all putative pyramidal neurons, which were identified with the following criteria: a triangular or pyramidal soma, the presence of a single apical dendrite extending in a similar orientation to other pyramidal cells in the same imaging session, and the presence of basal dendrites smaller in diameter than the apical dendrite. In a subset of experiments in which apical dendrites could not be clearly distinguished in the imaging plane, z-stack images were taken at the end of the imaging session, averaged, and referenced to visualize apical dendrites. After the identification of fluorescently tagged interneurons and pyramidal neurons, relevant putative interneurons were morphologically identified and compiled (e.g. putative VIP and SST neurons in PV-tagged mice). Putative interneurons were identified by the presence of an elongated soma and bipolar dendritic arbors (likely VIP), round somas smaller in diameter than pyramidal cells, and neurons with multiple perisomatic dendrites (usually highly branched; likely PV). The morphology of untagged interneurons was visually compared to the morphology of tagged interneurons in relevant mice (e.g.

morphologically identified interneurons in B6 PV[cre] mice were compared to fluorescently tagged interneurons in Sst-IRES-Cre and Vip-IRES-Cre mice). The average (11.07 μm, SD 1.64 μm) and maximum (18.14 μm, SD 2.74 μm) soma diameter for morphologically identified pyramidal neurons (n=3358) were significantly higher than in morphologically identified interneurons in the same mice (n=3522; average: 9.90 μm, SD 1.96; maximum: 16.37 μm, SD: 3.39 μm; p<0.001 for both values, unpaired t-test) and tagged VIP, SST, and PV neurons (n=543; average: 8.96 μm, SD 1.87 μm; maximum: 14.98 μm, SD 3.51 μm; p<0.001 for both values, unpaired t-test).

## Intrinsic signal imaging
### Image acquisition
Mice underwent the same surgical procedures described in *Two-Photon calcium imaging: Cranial window surgeries for imaging of barrel cortex neurons*. To facilitate manipulation of single whiskers, mice were lightly anesthetized with 10 mg/kg ketamine and placed on a stereotaxic frame. Using a custom GUI written in MATLAB, small vessels were visualized under a green LED (525 nm wavelength) with a high-speed camera (Genie Nano GigE, Teledyne DALSA, Waterloo, Ontario, Canada). Focus in the z-plane was adjusted approximately 250 μm below the putative pial surface before taking a reference image of the vasculature. For measurement of intrinsic signals, the field of view was illuminated with red light (625 nm). An Arduino Uno and MATLAB were used to trigger trial onset, the device used to mechanically deflect whiskers (a glass capillary tube attached to a rotary motor), a blue (470 nm) LED to stimulate climbing fibers, and recording of intrinsic signals at 30 Hz. Trials consisted of a 1 s baseline period, followed by a 1 s test pulse in which whiskers contralateral to the optical window were mechanically deflected at 8 Hz (RWS). This was repeated with an inter-trial interval of 18 s for 30 min prior to plasticity induction, after which whiskers were stimulated at 8 Hz for 5 min. Test pulses were delivered for 40 min after plasticity induction. For RWS+CF and control experiments, 50 ms blue light pulses were delivered at 1 Hz during plasticity induction with or without 8 Hz co-stimulation of whiskers, respectively.

### Image processing
Evoked signals were calculated similarly to *Vasquez et al., 2023*. Briefly, images were smoothed with a Gaussian filter and spatially downsampled by a factor of four. For each trial, an average baseline reflectance image (R0) was generated by calculating the average across the 1 s (30 frame) period before stimulus onset. Post-stimulus reflectance images were temporally averaged across 200 ms bins for 600 ms total, starting 400 ms after stimulus onset. The change in reflectance, ΔR/R, was calculated by subtracting R0 from each post-stimulus image and dividing the result by R0. Each of the three bins (400–600 ms, 600–800 ms, and 800 ms-1 s post stimulus onset) were averaged across a minimum of 30 trials and finally summed to yield a single total stimulus-evoked ΔR/R image. The reference image taken with green light illumination was used to draw a region of interest around the cranial window. Pixels outside this ROI mask were set to zero. Binary images were generated by calculating z-scores of the ΔR/R image and thresholding values below a z-score of −1.5 (note negative changes in reflectance indicate an increase in the presence of deoxyhemoglobin caused by increased oxygen consumption by active neurons). Binarized images were overlaid onto reference images. To calculate the areal extent of single whisker-evoked responses, clusters of active pixels were identified by applying a median filter with a 5×5-pixel neighborhood size, and the number of pixels above threshold was calculated. To reduce the impact of residual noise (i.e. detected pixels outside the area reliably activated by a single whisker), values were normalized by the area of the cranial window.

## Two-photon image processing
### Processing of cerebellar data
Processing of Purkinje cell calcium traces was conducted as previously described (*Busch and Hansel, 2023*; *Lin et al., 2024*). Individual trials from each session were concatenated along the z-axis, representing a time series across the entire recording session, to ensure consistency among trials after motion correction. The concatenated file underwent motion correction using a custom MATLAB script (MATLAB R2017b) based on whole frame cross-correlation (provided by the lab of Mark Sheffield, University of Chicago). Cellular regions of interest were manually selected in Fiji based on the average field of view across trials. Trials were then separated for further processing. In a subset of

trials, optogenetic stimulus artifacts were visible in a portion of the field of view in 1–3 nonconsecutive frames, depending on the stimulus duration. These frames were manually identified and excluded using Fiji before further processing.

Using custom MATLAB script (version R2022a), ΔF/F values were calculated from the raw trace using the following equation: $(F_t - F_0)/F_0$, where $F_t$ represented the raw calcium intensity of the time series, and the baseline fluorescence ($F_0$) was set as the 20th percentile of the fluorescence trace from each trial. ΔF/F values were then subjected to low-pass filtering using a five-frame moving window smoothing function. Finally, traces were normalized to a pre-stimulus baseline period by subtracting the average ΔF/F value from the 5-frame period prior to stimulus onset from the entire trace. To visualize population average responses to optogenetic stimulation of climbing fibers, calcium signals from each ROI were first trial-averaged, followed by concatenation and averaging of the signal across cells and mice. To calculate maximum amplitude of responses in the evoked period, the maximum in a window 0–700 ms after stimulus onset was calculated for each ROI's trial-averaged trace, then concatenated across cells and mice. Area under the curve (AUC) was calculated similarly by integrating the ΔF/F values over the same window. To identify spontaneous events, frame indices in which the first and second derivatives of the ΔF/F signal reached 0.04 were identified ($F^{idx}$). Peaks were detected using a peak prominence threshold of 0.1 ΔF/F, with spontaneous events defined as peaks occurring within 200 ms of $F^{idx}$. The onset of each spontaneous event was determined as the first frame in which the first derivative of the ΔF/F dropped below 0.05 (working backward from frame index of the detected peak). A 1 s window following event onset was extracted. Finally, after averaging all detected spontaneous events within ROIs, spontaneous events were compiled across neurons to generate the population average trace.

## Processing of S1 data

Motion correction and ROI curation were performed as above (see *Processing of cerebellar data* and *Two-photon calcium imaging: Characterization of neurons*). Residual optogenetic stimulus artifacts were observed in a single frame in a subset of trials, typically at or immediately after stimulus offset. These artifacts were detected using the *findpeaks* function in MATLAB and corrected by replacing the artifact frame's value with a linear interpolation of the two adjacent frames before further processing. Background subtraction was performed by calculating the bottom 1st percentile of the raw fluorescence trace collected from an ROI that did not contain cell bodies or neuropil, then subtracting this value from all trials. Smoothing, expression of raw signals as ΔF/F, calculation of baseline noise (σ), and calculation of the signal-to-noise ratio (SNR) were performed as in *Ayaz et al., 2019*. Smoothing was performed with a 51-point 1st-order Savitsky-Golay filter, followed by calculation of the relative percent change of fluorescence $((F_t - F_0)/F_0)$ using the 1st percentile of the smoothed trace as $F_0$. Given background subtraction prior to calculation of $F_0$ may generate negative $F_0$ values and subsequent signal inversion, ROIs with $F_0 < 0$ were rejected from further analyses (44/7423 cells). To calculate σ for each neuron, a 5 s (155 frame) sliding window was first used to calculate the standard deviation of the fluorescence change during each 5 s period within a session. After compiling all values, the 1st percentile value was taken as σ. For each neuron, the SNR was defined as the 95th percentile of ΔF/F signals recorded across all trials divided by the baseline noise.

## Evoked event detection

To detect whisker–evoked events, we first applied OASIS deconvolution to the ΔF/F traces using an event detection threshold of 3σ to generate a binary trace containing event times. An autoregressive model with order p=1 was used to ensure the number of events ('spikes') for each calcium transient varied with the amplitude of the calcium transient as previously observed (*Theis et al., 2016*). Next, for each frame, the sum of all events was calculated and divided by the number of trials to determine the proportion of trials with an event in each frame ($P^{event}$). $P^{event}$ was averaged across frames, and consecutive frames with $P^{event}$ three standard deviations above the mean were selected. This generated an evoked event window of 387 ms (*Figure 1—figure supplement 2B*). Finally, for each neuron, the calcium traces for all trials containing an event within the evoked event window were compiled for further analysis. To visualize population average traces for all experiments, evoked calcium signals for each neuron were first averaged across trials, followed by averaging across neurons. For visualizations between the same neural populations in different conditions, only those cells that responded to both

conditions, i.e., having an evoked event in each condition, were included (following the definition of 'Persistent cells' used in *Williams et al., 2025*). Shaded regions in population average traces represent SEM.

## Calculation of amplitude and area under the curve in S1 recordings

For all calculations of AUC in S1 recordings, the ΔF/F signal was integrated over a 700 ms window after stimulus onset unless otherwise specified (details below). Note that AUC is equal to the average response in the 700 ms window, multiplied by the 700 ms window duration (thus, AUC is directly proportional to the mean). We choose to report AUC, a descriptive statistic, rather than the mean within this 700 ms window.

To quantify plasticity phenomena caused by RWS or RWS+CF, all evoked events measured within 50 min prior to plasticity induction ('Pre') or 60 min after plasticity induction ('Post'") were extracted for each cell and trial-averaged before calculating AUC. Only the neurons with evoked events both pre- and post-plasticity induction were included in subsequent analyses. AUC were averaged both across all neurons or within animals to perform statistics across neurons or mice (see *Statistics and quantifications*). To visualize the duration of plasticity effects, evoked events were binned into 10 min epochs and trial-averaged within cells, followed by calculation of the AUC and averaging across neurons. To verify the robustness of plasticity effects, AUC and time course measurements were also analyzed across all cells and trials (i.e. those that did not have stimulus-evoked events; *Figure 1— figure supplement 2*) and in the absence of animal movement (*Figure 1—figure supplement 3*; see *Determination of active and rest trials*). To quantify plasticity phenomena in DREADD manipulation experiments, response amplitude was calculated in addition to AUC before and after plasticity induction. Response amplitude was taken as the maximum ΔF/F value in the same 700 ms window used for AUC calculations.

To quantify the immediate effect of climbing fiber activation on basic whisker responses in S1 cortex, AUC was calculated similarly for whisker stimulation (W) or whisker and CF coactivation (W+CF) trials. An additional measurement was taken 650–850 ms after stimulus onset for pyramidal neurons and VIP interneurons, as the effect of CF modulation was present in the later phase of the calcium response. Note that while W and W+CF test pulses were delivered prior to DCZ administration in DREADD experiments to verify functional opsin expression, these mice were excluded from this analysis due to an insufficient number of trials in each animal.

To verify that hM4D(Gi) and hM3D(Gq) expression in SST neurons indeed suppressed or increased their activity, respectively, AUC was calculated before and after DCZ administration in a window 1750 ms after stimulus onset. This larger window was used to capture the significantly increased duration of whisker-evoked calcium responses in SST neurons expressing hM3D(Gq). To verify that hM4D(Gi) and hM3D(Gq) expression in VIP neurons indeed suppressed or increased their activity, respectively, AUC was calculated before and after DCZ administration in a window 650–850 ms after stimulus onset. Given VIP neurons are modulated by climbing fiber stimulation in this phase of the calcium response (*Figure 2G*), this later window was chosen to ensure DREADD manipulation of VIP neurons suppressed or enhanced VIP responses in the physiologically relevant response period.

## Determination of active and rest trials

Activity of mice was monitored at 30.98 Hz using a DALSA M640 CCD camera (Teledyne Technologies, Thousand Oaks, CA, USA) focused on the animal's face and a portion of the running wheel. To determine the frames in which mice were active, custom MATLAB script was used to generate a binary trace that indicated the frames in which movement between frames occurred (based on a comparison of pixel values between consecutive frames). For each recording session, movement traces and corresponding videos from ten randomly selected trials were manually inspected to ensure all frames in which mice were whisking or running were indicated. Note that whisking occurred independently of running, but running seldom occurred without whisking. Rest trials were defined as any trial with no movement detected at least 400 ms before and after stimulus onset – longer than the evoked event window, and the same as the window chosen in *Ayaz et al., 2019* to distinguish rest from running onset. All other trials were considered active trials.

## Statistics and quantifications

All statistics and quantifications were performed using MATLAB (version R2022a). To assess normality, we applied the Lilliefors test to each dataset before performing statistical calculations. For paired samples, a paired t-test was used when data were normally distributed, and a Wilcoxon signed-rank test was used for non-normal data. For unpaired comparisons, an unpaired t-test was used for normally distributed data, a Welch's t-test for normally distributed data with unequal variances, and a Wilcoxon rank-sum test for non-normal distributions. Categorical variables were compared using the Chi-squared test. Two-way ANOVA was used for multi-group comparisons (i.e. to determine whether plasticity effects across time were significantly different between groups). To perform statistics across all animals, average responses were first normalized by dividing the response in the relevant experimental condition by the baseline response. A one-sample t-test was then performed against a standard value of 1. Statistical significance was set to $p < 0.05$, with p values presented in figures as follows: $p < 0.05$: *; $p < 0.01$: **; $p < 0.001$: ***; $p > 0.05$: not significant (n.s.). Bar graphs represent mean ± SEM. Box plots contain the interquartile range ($25^{th}$ and $75^{th}$ percentiles), median, and range.

## Acknowledgements

We thank Drs. M Brecht (Humboldt University, Berlin) as well as AM Oswald and SM Sherman (both University of Chicago) for invaluable feedback on the manuscript. We thank SR Postlewaite and current and former Hansel lab members SE Busch, D Huang, and TF Lin for insightful discussions and comments on the manuscript. This work was supported by the National Institutes of Health (NINDS) grants R21NS136954 (to CH) and NS094184 (to KPK in the lab of SM Sherman).

## Additional information

### Funding

| Funder | Grant reference number | Author |
| --- | --- | --- |
| National Institute of Neurological Disorders and Stroke | R21NS136954 | Christian Hansel |
| National Institute of Neurological Disorders and Stroke | NS094184 | Kevin P Koster |

The funders had no role in study design, data collection and interpretation, or the decision to submit the work for publication.

### Author contributions

Abby Silbaugh, Conceptualization, Data curation, Software, Formal analysis, Investigation, Visualization, Methodology, Writing – original draft, Writing – review and editing; Kevin P Koster, Data curation, Software, Formal analysis, Investigation, Methodology, Writing – review and editing, Funding acquisition, Visualization; Christian Hansel, Conceptualization, Resources, Supervision, Funding acquisition, Validation, Visualization, Writing – original draft, Project administration, Writing – review and editing

### Author ORCIDs

Abby Silbaugh ⓘ https://orcid.org/0000-0002-5606-6485
Kevin P Koster ⓘ https://orcid.org/0000-0003-2935-3427
Christian Hansel ⓘ https://orcid.org/0000-0001-5750-7097

### Ethics

All experiments and surgical procedures were performed in accordance with the University of Chicago Animal Care and Use Committee guidelines (protocol ACUP 72496).

Reviewer #1 (Public review): https://doi.org/10.7554/eLife.109183.3.sa1
Reviewer #2 (Public review): https://doi.org/10.7554/eLife.109183.3.sa2
Reviewer #3 (Public review): https://doi.org/10.7554/eLife.109183.3.sa3
Author response https://doi.org/10.7554/eLife.109183.3.sa4

## Additional files

### Supplementary files
MDAR checklist

### Data availability
All data and code have been deposited on Github: https://github.com/abbysilbaugh/climbingfiber (copy archived at *Silbaugh, 2025*).

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
