## [Editor Report · eLife Assessment]

This study presents a **fundamental** discovery of how cerebellar climbing fibers modulate plastic changes in the somatosensory cortex by identifying both the responsible cortical circuit and the anatomical pathways. The evidence supporting the conclusions is **convincing** and well supported by modern neuroscience methodologies. Overall, this work represents a significant contribution that will be of broad interest to neuroscientists, especially those studying the long-distance cerebellar influence on non-motor brain functions.

---

## [Referee Report · Reviewer #1 (Public review)]

Summary:

Silbaugh, Koster and Hansel investigated how the cerebellar climbing fiber (CF) signals influence neuronal activity and plasticity in mouse primary somatosensory (S1) cortex. They found that optogenetic activation of CFs in the cerebellum modulates responses of cortical neurons to whisker stimulation in a cell-type-specific manner and suppresses potentiation of layer 2/3 pyramidal neurons induced by repeated whisker stimulation. This suppression of plasticity by CF activation is mediated through modulation of VIP- and SST-positive interneurons. Using transsynaptic tracing and chemogenetic approaches, the authors identified a pathway from the cerebellum through the zona incerta and the thalamic posterior medial (POm) nucleus to the S1 cortex, which underlies this functional modulation.

The authors have addressed all the necessary points.

---

## [Referee Report · Reviewer #2 (Public review)]

Summary:

The authors examined long-distance influence of climbing fiber (CF) signaling in the somatosensory cortex by manipulating whiskers through stimulation. Also, they examined CF signaling using two-photon imaging and mapped projections from the cerebellum to somatosensory cortex using transsynaptic tracing. As a final manipulation, they used chemogenetics to perturb parvalbumin positive neurons in the zona incerta and recorded from climbing fibers.

Strengths:

There are several strengths to this paper. The recordings were carefully performed and AAVs used were selective and specific for the cell-types and pathways being analyzed. In addition, the authors used multiple approaches that support climbing fiber pathways to distal regions of the brain. This work will impact the field and describes nice methods to target difficult to reach brain regions, such as the inferior olive.

No weaknesses noted.

---

## [Referee Report · Reviewer #3 (Public review)]

Summary:

The authors developed an interesting novel paradigm to probe the effects of cerebellar climbing fiber activation on short-term adaptation of somatosensory neocortical activity during repetitive whisker stimulation. Normally, RWS potentiated whisker responses in pyramidal cells and weakly suppressed them in interneruons, lasting for at least 1h. Crusii Optogenetic climbing fiber activation during RWS reduced or inverted these adaptive changes. This effect was generally mimicked or blocked with chemogenetic SST or VIP activation/suppression as predicted based on their "sign" in the circuit.

Strengths:

The central finding about CF modulation of S1 response adaptation is interesting, important, and convincing, and provides a jumping-off point for the field to start to think carefully about cerebellar modulation of neocortical plasticity.

Weaknesses:

The SST and VIP results appeared slightly weaker statistically, but I do not personally think this detracts from the importance of the initial finding (if there are multiple underlying mechanisms, modulating one may reproduce only a fraction of the effect size). I found the suggestion that zona incerta may be responsible for the cerebellar effects on S1 to be a more speculative result (it is not so easy with existing technology to effectively modulate this type of polysynaptic pathway), but this may be an interesting topic for the authors to follow up on in more detail in the future.

Comments on revisions:

The authors have appropriately addressed my comments.

---

## [Author Response]

The following is the authors’ response to the original reviews

**Public Reviews:**

**Reviewer #1 (Public review):**
Summary:Silbaugh, Koster, and Hansel investigated how the cerebellar climbing fiber (CF) signals influence neuronal activity and plasticity in mouse primary somatosensory (S1) cortex. They found that optogenetic activation of CFs in the cerebellum modulates responses of cortical neurons to whisker stimulation in a cell-type-specific manner and suppresses potentiation of layer 2/3 pyramidal neurons induced by repeated whisker stimulation. This suppression of plasticity by CF activation is mediated through modulation of VIP- and SST-positive interneurons. Using transsynaptic tracing and chemogenetic approaches, the authors identified a pathway from the cerebellum through the zona incerta and the thalamic posterior medial (POm) nucleus to the S1 cortex, which underlies this functional modulation.Strengths:This study employed a combination of modern neuroscientific techniques, including two-photon imaging, opto- and chemo-genetic approaches, and transsynaptic tracing. The experiments were thoroughly conducted, and the results were clearly and systematically described. The interplay between the cerebellum and other brain regions - and its functional implications - is one of the major topics in this field. This study provides solid evidence for an instructive role of the cerebellum in experience-dependent plasticity in the S1 cortex.Weaknesses:There may be some methodological limitations, and the physiological relevance of the CFinduced plasticity modulation in the S1 cortex remains unclear. In particular, it has not been elucidated how CF activity influences the firing patterns of downstream neurons along the pathway to the S1 cortex during stimulation.

Our study addresses the important question of whether CF signaling can influence the activity and plasticity of neurons outside the olivocerebellar system, and further identifies the mechanism through which this indeed occurs. We provide a detailed description of the involvement of specific neuron subtypes and how they are modulated by climbing fiber activation to impact S1 plasticity. We also identify at least one critical pathway from the cerebellar output to the S1 circuit. It is indeed correct that we did not investigate how the specific firing patterns of all of these downstream neurons are affected, or the natural behaviors in which this mechanism is involved. Now that it is established that CF signaling can impact activity and plasticity outside the olivocerebellar system -- and even in the primary somatosensory cortex -- these questions will be important to further investigate in future studies.

(1) Optogenetic stimulation may have activated a large population of CFs synchronously, potentially leading to strong suppression followed by massive activation in numerous cerebellar nuclear (CN) neurons. Given that there is no quantitative estimation of the stimulated area or number of activated CFs, observed effects are difficult to interpret directly. The authors should at least provide the basic stimulation parameters (coordinates of stim location, power density, spot size, estimated number of Purkinje cells included, etc.).

As discussed in the paper, we indeed expect that synchronous CF activation is needed to allow for an effect on S1 circuits under natural or optogenetic activation conditions. The basic optogenetic stimulation parameters (also stated in the methods) are as follows: 470 nm LED; Ø200 µm core, 0.39 NA rotary joint patch cable; absolute power output of 2.5 mW; spot size at the surface of the cortex 0.6 mm; estimated power density 8 mW/mm2. A serious estimate of the number of Purkinje cells that are activated is difficult to provide, in particular as ‘activation’ would refer to climbing fiber inputs, not Purkinje cells directly.

(2) There are CF collaterals directly innervating CN (PMID:10982464). Therefore, antidromic spikes induced by optogenetic stimulation may directly activate CN neurons. On the other hand, a previous study reported that CN neurons exhibit only weak responses to CF collateral inputs (PMID: 27047344). The authors should discuss these possibilities and the potential influence of CF collaterals on the interpretation of the results.

A direct activation of CN neurons by antidromic spikes in CF collaterals cannot be ruled out. However, we believe that this effect will not be substantial. The activation of the multi-synaptic pathway that we describe in this study is more likely to require a strong nudge as resulting from synchronized Purkinje cell input and subsequent rebound activation in CN neurons (PMID: 22198670), rather than small-amplitude input provided by CF collaterals (PMID: 27047344). A requirement for CF/PC synchronization would also set a threshold for activation of this suppressive pathway.

(3) The rationale behind the plasticity induction protocol for RWS+CF (50 ms light pulses at 1 Hz during 5 min of RWS, with a 45 ms delay relative to the onset of whisker stimulation) is unclear.a) The authors state that 1 Hz was chosen to match the spontaneous CF firing rate (line 107); however, they also introduced a delay to mimic the CF response to whisker stimulation (line 108). This is confusing, and requires further clarification, specifically, whether the protocol was designed to reproduce spontaneous or sensory-evoked CF activity.

This protocol was designed to mimic sensory-evoked CF activity as reported in Bosman et al (J. Physiol. 588, 2010; PMID: 20724365).

b) Was the timing of delivering light pulses constant or random? Given the stochastic nature of CF firing, randomly timed light pulses with an average rate of 1Hz would be more physiologically relevant. At the very least, the authors should provide a clear explanation of how the stimulation timing was implemented.

Light pulses were delivered at a constant 1 Hz. Our goal was to isolate synchrony as the variable distinguishing sensory-evoked from spontaneous CF activity; additionally varying stochasticity, rate, or amplitude would have confounded this. Future studies could explore how these additional parameters shape S1 responses.

(4) CF activation modulates inhibitory interneurons in the S1 cortex (Figure 2): responses of interneurons in S1 to whisker stimulation were enhanced upon CF coactivation (Figure 2C), and these neurons were predominantly SST- and PV-positive interneurons (Figure 2H, I). In contrast, VIP-positive neurons were suppressed only in the late time window of 650-850 ms (Figure 2G). If the authors' hypothesis-that the activity of VIP neurons regulates SST- and PVneuron activity during RWS+CF-is correct, then the activity of SST- and PV-neurons should also be increased during this late time window. The authors should clarify whether such temporal dynamics were observed or could be inferred from their data.

Yes, we see a significant activity increase in PV neurons in this late time window (see updates to Data S2). Activity was also increased in SST neurons, though this did not reach statistical significance (Data S2). One reason might be that – given the small effect size overall – such an effect would only be seen in paired recordings. Chemogenetic activity modulation in VIP neurons, which provides a more crude test, shows, however, that SST- and PV-positive interneurons are indeed regulated via inhibition from VIP-positive interneurons (Fig. 5).

(5) Transsynaptic tracing from CN nicely identified zona incerta (ZI) neurons and their axon terminals in both POm and S1 (Figure 6 and Figure S7).a) Which part of the CN (medial, interposed, or lateral) is involved in this pathway is unclear.

We used a dual-injection transsynaptic tracing approach to specifically label the outputs of ZI neurons that receive input from the deep cerebellar nuclei. The anterograde viral vector injected into the CN is unlabeled (no fluorophore) and therefore, it is not possible to reliably assess the extent of viral spread in those experiments as performed. However, we have previously performed similar injections into the deep cerebellar nuclei and post hoc histology suggest all three nuclei will have at least some viral expression (Koster and Sherman, 2024). Due to size and injection location, we will mostly have reached the lateral (dentate) nuclei, but cannot exclude partial transsynaptic tracing from the interposed and medial nuclei.

b) Were the electrophysiological properties of these ZI neurons consistent with those of PV neurons?

Although most recorded cells demonstrated electrophysiological properties consistent with PV+ interneurons in other brain regions (i.e. fast spiking, narrow spike width, non-adapting; see Tremblay et al., 2016), interneuron subtypes in the ZI have been incompletely characterized, with SST+ cells showing similar features to those typically associated with PV+ cells (if interested, compare Fig. 4 in DOI: 10.1126/sciadv.abf6709 vs. Fig. S10 in https://doi.org/10.1016/j.neuron.2020.04.027). Therefore, we did not attempt to delineate cell identity based on these characteristics.

c) There appears to be a considerable number of axons of these ZI neurons projecting to the S1 cortex (Figure S7C). Would it be possible to estimate the relative density of axons projecting to the POm versus those projecting to S1? In addition, the authors should discuss the potential functional role of this direct pathway from the ZI to the S1 cortex.

An absolute quantification is difficult to provide based on the images that we obtained. However, any crude estimate would indicate the relative density of projections to POm is higher than the density of projections to S1 (this is apparent from the images themselves). While the anatomical and functional connections from POm to S1 have been described in detail (Audette et al., 2018), this is not the case for the direct projections to ZI. A direct ZI to S1 projection would potentially involve a different recruitment of neurons in the S1 circuit. Any discussion on the specific consequences of the activation of this direct pathway would be purely speculative.

**Reviewer #2 (Public review):**
Summary:The authors examined long-distance influence of climbing fiber (CF) signaling in the somatosensory cortex by manipulating whiskers through stimulation. Also, they examined CF signaling using two-photon imaging and mapped projections from the cerebellum to the somatosensory cortex using transsynaptic tracing. As a final manipulation, they used chemogenetics to perturb parvalbumin-positive neurons in the zona incerta and recorded from climbing fibers.Strengths:There are several strengths to this paper. The recordings were carefully performed, and AAVs used were selective and specific for the cell types and pathways being analyzed. In addition, the authors used multiple approaches that support climbing fiber pathways to distal regions of the brain. This work will impact the field and describes nice methods to target difficult-to-reach brain regions, such as the inferior olive.Weaknesses:There are some details in the methods that could be explained further. The discussion was very short and could connect the findings in a broader way.

In the revised manuscript, we provide more methodological details, as requested. We provided as simple as possible explanations in the discussion, so as not to bias further investigations into this novel phenomenon. In particular, we avoid an extended discussion of the gating effect of CF activity on S1 plasticity. While this is the effect on plasticity specifically observed here, we believe that the consequences of CF signaling on S1 activity may entirely depend on the contexts in which CF signals are naturally recruited, the ongoing activity of other brain regions, and behavioral state. Our key finding is that such modulation of neocortical plasticity can occur. How CF signaling controls plasticity of the neocortex in all contexts remains unknown, but needs to be thoughtfully tested in the future.

**Reviewer #3 (Public review):**
Summary:The authors developed an interesting novel paradigm to probe the effects of cerebellar climbing fiber activation on short-term adaptation of somatosensory neocortical activity during repetitive whisker stimulation. Normally, RWS potentiated whisker responses in pyramidal cells and weakly suppressed them in interneurons, lasting for at least 1h. Crusii Optogenetic climbing fiber activation during RWS reduced or inverted these adaptive changes. This effect was generally mimicked or blocked with chemogenetic SST or VIP activation/suppression as predicted based on their "sign" in the circuit.Strengths:The central finding about CF modulation of S1 response adaptation is interesting, important, and convincing, and provides a jumping-off point for the field to start to think carefully about cerebellar modulation of neocortical plasticity.Weaknesses:The SST and VIP results appeared slightly weaker statistically, but I do not personally think this detracts from the importance of the initial finding (if there are multiple underlying mechanisms, modulating one may reproduce only a fraction of the effect size). I found the suggestion that zona incerta may be responsible for the cerebellar effects on S1 to be a more speculative result (it is not so easy with existing technology to effectively modulate this type of polysynaptic pathway), but this may be an interesting topic for the authors to follow up on in more detail in the future.

Our interpretation of the anatomical and physiological findings is that a pathway via the ZI is indeed critical for the observed effects. This pathway also represents perhaps the most direct pathway (i.e. least number of synapses connecting the cerebellar nuclei to S1). However, several other direct and indirect pathways are plausible as well and we expect distinct activation requirements and consequences for neurons in the S1 circuit. These are indeed interesting topics for future investigation.

**Recommendations for the authors:**

**Reviewer #1 (Recommendations for the authors):**
(1) Line 77: "CF transients" is not a standard or widely recognized term. Please use a more precise expression, such as "CF-induced calcium transients."

We now avoid the use of the term “CF transients” and replaced it with “CF-induced calcium transients.”

(2) Titer of AAVs injected should be provided.

AAV titers have been included in an additional data table (Data S9).

(3) Several citations to the figures are incorrect (for example, "Supplementary Data 2a (Line 398)" does not exist).

We apologize for the mistakes in this version of the article. Incorrect citations to the figures have been corrected.

(4) Line 627-628: "The tip of the patch cable was centered over Crus II in all optogenetic stimulation experiments." The stereotaxic coordinate of the tip position should be provided.

The stereotaxic coordinate of the tip position has been provided in the methods.

(5) Line 629: "Blue light pulses were delivered with a 470 nm Fiber-Coupled LED (Thorlabs catalog: M470F3)." The size of the light stim and estimated power density (W/mm^2) at the surface of the cortex should be provided.

The spot size and estimated power density at the surface of the cortex has been provided in the methods.

(6) Line 702-706: References for DCZ should be cited.

We now cited Nagai et al, Nat. Neurosci. 23 (2020) as the original reference.

(7) Two-photon image processing (Line 807-809): The rationale for normalizing ∆F/F traces to a pre-stimulus baseline is unclear because ∆F/F is, by definition, already normalized to baseline fluorescence: (Ft-F0)/F0. The authors should clarify why this additional normalization step was necessary and how it affected the interpretation of the data.

A single baseline fluorescence value (F₀) was computed for each neuron across the entire recording session, which lasted ~120-minutes. However, some S1 neurons exhibit fluctuations in baseline fluorescence over time—often related to locomotive activity or spontaneous network oscillations—which can obscure stimulus-evoked changes. To isolate fluorescence changes specifically attributable to whisker stimulation, we normalized each ∆F/F trace to the prestimulus baseline for that trial. This additional normalization allowed us to quantify potentiation or depression of sensory responses themselves, independently of spontaneous oscillations or locomotion-related changes in the ongoing neural activity.

**Reviewer #2 (Recommendations for the authors):**
(1) Did the climbing fiber stimulation for Figure 1 result in any changes to motor activity? Can you make any additional comments on other behaviors that were observed during these manipulations?

Acute CF stimulation did not cause any changes in locomotive or whisking activity. The CF stimulation also did not influence the overall level of locomotion or whisking during plasticity induction.

(2) Figure 3B and F- it is very difficult to see the SST+ neurons. Can this be enhanced?

We linearly adjusted the brightness and contrast for the bottom images in Figure 3B and F to improve visualization of SST+ neurons. Note the expression of both hM3D(Gq) and hM4D(Gi) in SST+ neurons is sparse, which was necessary to avoid off-target effects.

(3) Can you be more specific about the subregions of cerebellar nuclei and cell types that are targeted in the tracing studies? Discussions of the cerebellar nuclei subregions are missing and would be interesting, as others have shown discrete pathways between cerebellar nuclei subregions and long-distance projections.

See our response to comment 5a from Reviewer 1 (copied again here): we used a dual-injection transsynaptic tracing approach to specifically label the outputs of ZI neurons that receive input from the deep cerebellar nuclei. The anterograde viral vector injected into the CN is unlabeled (no fluorophone) and therefore, it is not possible to reliably assess the extent of viral spread in those experiments as performed. However, we have previously performed similar injections into the deep cerebellar nuclei and post hoc histology suggest all three nuclei will have at least some viral expression (Koster and Sherman, 2024). Due to size and injection location, we will mostly have reached the lateral (dentate) nuclei, but cannot exclude partial transsynaptic tracing from the interposed and medial nuclei.

It would indeed be interesting to further investigate the effect of CFs residing in different cerebellar lobules, which preferentially target different cerebellar nuclei, on targets of these nuclei.

(4) Did you see any connection to the ventral tegmental area? Can you comment on whether dopamine pathways are influenced by CF and in your manipulations?

We did not specifically look at these pathways and thus are not able to comment on this.

(5) These are intensive surgeries, do you think glia could have influenced any results?

This was not tested and seems unlikely, but we cannot exclude such possibility.

(6) It is unclear in the methods how long animals were recorded for in each experiment. Can you add more detail?

Additional detail was added to the methods. Recordings for all experimental configurations did not last more than 120 minutes in total. All data were analyzed across identical time windows for each experiment.

(7) In the methods it was mentioned that recording length can differ between animals. Can this influence the results, and if so, how was that controlled for?

There was a variance in recording length within experimental groups, but no systematic difference between groups.

(8) I do not see any mention of animal sex throughout this manuscript. If animals were mixed groups, were sex differences considered? Would it be expected that CF activity would be different in male and female mice?

As mentioned in the Methods (Animals), mice of either sex were used. No sex-dependent differences were observed.

(9) Transsynaptic tracing results of the zona incerta are very interesting. The zona incerta is highly understudied, but has been linked to feeding, locomotion, arousal, and novelty seeking. Do you think this pathway would explain some of the behavioral results found through other studies of cerebellar lobule perturbations? Some discussion of how this brain region would be important as a cerebellar connection in animal behavior would be interesting.

Since the multi-synaptic pathway from the cerebellum to S1 involves several brain regions with their own inputs and modulatory influences, it seems plausible to assume that behaviors controlled by these regions or affecting signaling pathways that regulate them would show some level of interaction. Our study does not address these interactions, but this will be an interesting question to be addressed in future work.

**Reviewer #3 (Recommendations for the authors):**
General comments on the data presentation:I'm not a huge fan of taking areas under curves ('AUC' throughout the study) when the integral of the quantity has no physical meaning - 'normalizing' the AUC (1I,L etc) is even stranger, because of course if you instead normalize the AUC by the # of data points, you literally just get the mean (which is probably what should be used instead).

Indeed, AUC is equal to the average response in the time window used, multiplied by the window duration (thus, AUC is directly proportional to the mean). We choose to report AUC, a descriptive statistic, rather than the mean within this window. In 1I and L, we normalize the AUC across animals, essentially removing the variability across animals in the ‘Pre’ condition for visualization. Note the significance of these comparisons are consistent whether or not we normalize to the ‘Pre’ condition (non-normalized RWS data in I shows a significant increase in PN activity, p = 0.0068, signrank test; non-normalized RWS+CF data in I shows a significant decrease in PN activity, p = 0.0135, paired t-test; non-normalized RWS data in L shows a significant decrease in IN activity, p <0.001, paired t-test; non-normalized RWS+CF data in L shows no significant change in IN activity, p = 0.7789, paired t-test).

I think unadorned bar charts are generally excluded from most journals now. Consider replacing these with something that shows the raw datapoints if not too many, or the distribution across points.

We have replaced bar charts with box plots and violin plots. We have avoided plotting individual data points due to the quantity of points.

In various places, the statistics produce various questionable outcomes that will draw unwanted reader scrutiny. Many of the examples below involve tiny differences in means with overlapping error bars that are "significant" or a few cases of nonoverlapping error bars that are "not significant." I think replacing the bar charts may help to resolve things here if we can see the whole distribution or the raw data points. As importantly, I think a big problem is that the statistical tests all seem to be nonparametric (they are ambiguously described in Table S3 as "Wilcoxon," which should be clarified, since there is an unpaired Wilcoxon test [rank sum] and a paired Wilcoxon test [sign rank]), and thus based on differences in the *median* whereas the bar charts are based on the *mean* (and SEM rather than MAD or IQR or other medianappropriate measure of spread). This should be fixed (either change the test or change the plots), which will hopefully allay many of the items below.

We thank the reviewer for this important point. As mentioned in the Statistics and quantification section, Wilcoxon signed rank tests were used for non-normal data. We have replaced the bar charts with box plots which show the IQR and median, which indeed allays may of the items below.

Here are some specific points on the statistics presentation:(1) 1G, the test says that following RWS+CF, the decrease in PN response is not significant. In 1I, the same data, but now over time, shows a highly significant decrease. This probably means that either the first test should be reconsidered (was this a paired comparison, which would "build in" the normalization subsequently used automatically?) or the second test should be reconsidered. It's especially strange because the n value in G, if based on cells, would seem to be ~50-times higher than that in I if based on mice.

In Figure 1G, the analysis tests whether individual pyramidal neurons significantly changed their responses before vs. after RWS+CF stimulation. This is a paired comparison at the single-cell level, and here indicates that the average per-neuron response did not reliably decrease after RWS+CF when comparing each cell’s pre- and post-values directly. In contrast, Figure 1I examines the same dataset analyzed across time bins using a two-way ANOVA, which tests for effects of time, group (RWS vs. RWS+CF), and their interaction. The analysis showed a significant group effect (p < 0.001), indicating that the overall level of activity across all time points differed between RWS and RWS+CF conditions. The difference in significance between these two analyses arises because the first test (Fig. 1G) assesses within-neuron changes (paired), whereas the second test (Fig. 1I) assesses overall population-level differences between groups over time (independent groups). Thus, the tests address related but distinct questions—one about per-cell response changes, the other about how activity differs across experimental conditions.

(2) 1J RWS+CF then shows a much smaller difference with overlapping error bars than the ns difference with nonoverlapping errors in 1G, but J gets three asterisks (same n-values).

Bar graphs have been replaced with box plots.

(3) 1K, it is very unclear what is under the asterisk could possibly be significant here, since the black and white dots overlap and trade places multiple times.

See response to point 1. A significant group effect will exist if the aggregate difference across all time bins exceeds within-group variability. The asterisk therefore reflects a statistically significant main group effect (RWS versus RWS+CF) rather than differences at any single time point. Note, however, the very small effect size here.

(4) 2B, 2G, 2H, 2I, 3G, 3H, 5C etc, again, significance with overlapping error bars, see suggestions above.

Bar graphs have been replaced with box plots.

(5) Time windows: e.g., L149-153 / 2B - this section reads weirdly. I think it would be less offputting to show a time-varying significance, if you want to make this point (there are various approaches to this floating around), or a decay rate, or something else.

Here, we wanted to understand the overall direction of influence of CFs on VIP activity. We find that CFs exert a suppressive effect on VIP activity, which is statistically significant in this later time window. The specific effect of CF modulation on the activity of S1 neurons across multiple time points will be described in more detail in future investigations.

(6) 4G, 6I, these asterisks again seem impossible (as currently presented).

Bar graphs have been replaced with box plots.

The writing is in generally ok shape, but needs tightening/clarifying:(1) L45 "mechanistic capacity" not clear.

We have simplified this term to “capacity.” We use the term here to express that the central question we pose is whether CF signals are able to impact S1 circuits. We demonstrate CF signals indeed influence S1 circuits and further describe the mechanism through which this occurs, but we do not yet know all of the natural conditions in which this may occur. We feel that “capacity” describes the question we pose -- and our findings -- very well.

(2) L48-58 there's a lot of material here, not clear how much is essential to the present study.

We would like to give an overview of the literature on instructive CF signaling within the cerebellum. Here, we feel it is important to describe how CFs supervise learning in the cerebellum via coincident activation of parallel fiber inputs and CF inputs. Our results demonstrate CFs have the capacity to supervise learning in the neocortex in a similar manner, as coincident CF activation with sensory input modulates plasticity of S1 neurons.

(3) L59 "has the capacity to" maybe just "can".

This has been adopted. We agree that “can” is a more straightforward way of saying “has the capacity to” here. In this sentence, “can” and “has the capacity to” both mean a general ability to do something, without explicit knowledge about the conditions of use.

(4) L61-62 some of this is circular "observation that CF regulates plasticity in S1..has consequences for plasticity in S1".

We now changed this to read “…consequences for input processing in S1.”

(5) L91 "already existing whisker input" although I get it, strictly speaking, not clear what this means.

This sentence has been reworded for clarity.

(6) L94 "this form of plasticity" what form?

Edited to read “sensory-evoked plasticity.”

(7) L119 should say "to test the".

This has been corrected.

(8) L120 should say "well-suited to measure receptive fields".

We agree; this wording has been adopted.

(9) L130 should say "optical imaging demonstrated that receptive field".

This has been adopted.

(10) L138, the disclaimer is helpful, but wouldn't it be less confusing to just pick a different set of terms? Response potentiation etc.

Perhaps, but we want to stress that components of LTP and LTD (traditionally tested using electrophysiological methods to specifically measure synaptic gain changes) can be optically measured as long as it is specified what is recorded.

(11) L140, this whole section is not very clear. What was the experiment? What was done and how?

The text in this section has been updated.

(12) L154, 156, 158, 160, 960, what is a "basic response"? Is this supposed to contrast with RWS? If so, I would just say "we measured the response to whisker stimulation without first performing RWS, and compared this to the whisker stimulation with simultaneous CF activation."

What we meant by “basic response” was the acute response of S1 neurons to a single 100 ms air puff. Here, we indeed measured the acute responses of S1 neurons to whisker stimulation (100 ms air puff) and compared them to whisker stimulation with simultaneous CF activation (100 ms air puff with a 50 ms light pulse; the light pulse was delayed 45 ms with respect to the air puff). This paragraph has been reworded for clarity.

(13) L156 "comprised of a majority" unclear. You mean most of the nonspecific IN group is either PV or SST?

Yes, that was meant here. This paragraph has been reworded for clarity.

(14) L165 tense. "are activated" "we tested" prob should be "were activated."

This sentence was reworded.

(15) L173 Not requesting additional experiments, but demonstrating that the effect is mimicked by directly activating SST or suppressing VIP questions the specificity of CF activation per se, versus presumably many other pathways upstream of the same mechanisms, which might be worth acknowledging in the text.

We indeed observe that directly activating SST or suppressing VIP neurons in S1 is sufficient to mediate the effect of CF activation on S1 pyramidal neurons, implicating SST and VIP neurons as the local effectors of CF signaling. In the text, we wrote “...the notion of sufficiency does not exclude potential effects of plasticity processes elsewhere that might well modulate effector activation in this context and others not yet tested.” Here, we mean that CFs are certainly not the only modulators of the inhibitory network in S1. One example we highlight in the discussion is that projections from M1 are known to modulate this disinhibitory VIP-to-SST-to-PN microcircuit in S1. We conclude from our chemogenetic manipulation experiments that CFs ultimately have the capacity to modulate S1 interneurons, which must occur indirectly (either through the thalamus or “upstream” regions as this reviewer points out). The fact that many other brain regions may also modulate the interneuron network in S1 -- or be modulated by CF activity themselves -- only expands the capacity of CFs to exert a variety of effects on S1 neurons in different contexts.

(16) L247 "induced ChR2" awkward.

We changed this to read “we expressed ChR2.”

(17) 6C, what are the three colors supposed to represent?

We apologize for the missing labels in this version of the manuscript. Figure 6C and the figure legend have been updated.